# A combinatorial code of neurexin-3 alternative splicing controls inhibitory synapses via a trans-synaptic dystroglycan signaling loop

**Justin H. Trotter** [1,3] ✉, **Cosmos Yuqi Wang**[1,3], **Peng Zhou**[1,3], **George Nakahara**[1] & **Thomas C. Südhof** [1,2] ✉

Disrupted synaptic inhibition is implicated in neuropsychiatric disorders, yet the molecular mechanisms that shape and sustain inhibitory synapses are poorly understood. Here, we show through rescue experiments performed using Neurexin-3 conditional knockout mice that alternative splicing at SS2 and SS4 regulates the release probability, but not the number, of inhibitory synapses in the olfactory bulb and prefrontal cortex independent of sex. Neurexin-3 splice variants that mediate Neurexin-3 binding to dystroglycan enable inhibitory synapse function, whereas splice variants that don't allow dystroglycan binding do not. Furthermore, a minimal Neurexin-3 protein that binds to dystroglycan fully sustains inhibitory synaptic function, indicating that trans-synaptic dystroglycan binding is necessary and sufficient for Neurexin-3 function in inhibitory synaptic transmission. Thus, Neurexin-3 enables a normal release probability at inhibitory synapses via a trans-synaptic feedback signaling loop consisting of presynaptic Neurexin-3 and postsynaptic dystroglycan.

Synapses are sophisticated intercellular junctions controlled by trans-synaptic signaling that is mediated, at least in part, by interactions between pre- and postsynaptic adhesion molecules. Among synaptic adhesion molecules (SAMs), neurexins stand out because of their central role in shaping the properties of synapses[1–4]. Neurexins perform a panoply of synaptic functions, ranging from mediating a normal neurotransmitter release probability[5–7] to controlling presynaptic $GABA_B$-receptors[8], regulating postsynaptic glutamate and $GABA_A$-receptors[9–12], and enabling postsynaptic NMDA-receptor-dependent LTP[13]. Recent observations uncovered multitudinous interactions of neurexins with diverse ligands that likely mediate the functions of neurexins. Indeed, trans-synaptic ligands for some of these functions were identified, as shown for the neurexin-dependent control of

glutamate receptors that are dictated by Cbln1/2-GluD1/2 complexes in the subiculum[11] or for neurexin-dependent LTP that requires *Nlgn1* in the hippocampal CA1 region[14]. The best-documented role of neurexins among their various functions probably consists of their regulation of the presynaptic release probability[5–7], but no candidate ligands were identified for that role, making it unclear how neurexins determine the release probability of a synapse. Even the question of whether presynaptic neurexins act directly cell-autonomously in the nerve terminal or operate indirectly via binding to a postsynaptic ligand that then signals back to the presynaptic release machinery has not been addressed[4].

Neurexins are transcribed from three genes (*Nrxn1-3*) in two principal forms, longer α-neurexins and shorter β-neurexins[15–17].

[1]Department of Molecular and Cellular Physiology, Stanford University School of Medicine, Stanford, CA 94305, USA. [2]Howard Hughes Medical Institute, Stanford University School of Medicine, Stanford, CA 94305, USA. [3]These authors contributed equally: Justin H. Trotter, Cosmos Yuqi Wang, Peng Zhou. ✉e-mail: justint3@stanford.edu; tcs1@stanford.edu

Neurexin mRNAs are extensively alternatively spliced at six canonical positions (SS1 to SS6)[18,19], whose use is highly regulated spatially and temporally[20,21]. The best-studied site of alternative splicing of neurexins, splice site #4 (SS4), regulates the binding of key ligands[1,4,22–25] and controls the postsynaptic levels of AMPA- (AMPARs) and NMDA-receptors (NMDARs)[10]. Another site of alternative splicing, SS2 that is only present in α-neurexins (Fig. 1a), has been found to modulate two neurexin-ligand interactions, binding to dystroglycan, a postsynaptic adhesion molecule, and to neurexophilins, a family of secreted cysteine-rich proteins[26–30]. Dystroglycan binds only to α-neurexins lacking an insert in either SS2 and/or in SS4, but not to α-neurexins with inserts in both SS2 and SS4, whereas neurexophilins

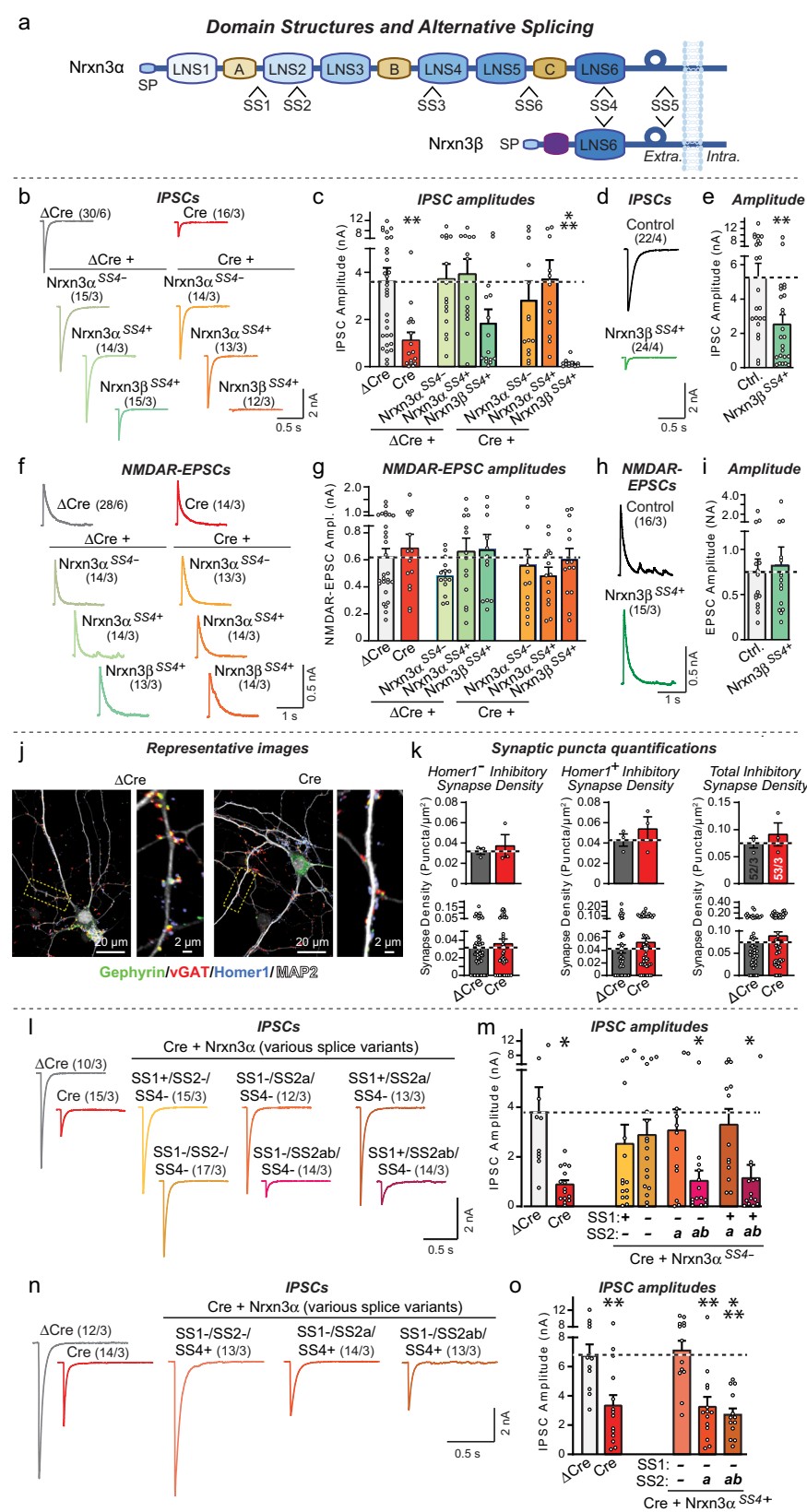

**Fig. 1 | Nrxn3α alternative splicing regulates inhibitory synapses in dissociated olfactory bulb cultures. a** Schematic of Nrxn3α/β structure (SP: signal peptide; LNS1-6: LNS domains; A, B, and C: EGF-like domains) and sites of alternative splicing (SS1-SS6). **b, c** Conditional *Nrxn3* deletion impairs inhibitory synaptic transmission monitored via evoked IPSC amplitudes (b, sample traces; c, IPSC amplitude summary graphs). **d, e** Nrxn3β overexpression suppresses evoked IPSCs in wild-type OB neurons (d, sample traces; e, summary graphs of IPSC amplitudes). **f, g** Conditional *Nrxn3* deletion has no effect on evoked EPSCs (f, sample traces of NMDAR-EPSCs; **g**, EPSC amplitude summary graphs). **h, i** Nrxn3β overexpression does not impair evoked EPSCs (h, sample traces of NMDAR-EPSCs; **i** EPSC amplitude summary graphs). **j, k** Conditional *Nrxn3* deletion does not alter inhibitory synapse numbers as analyzed by immunocytochemistry for presynaptic inhibitory (vGAT), post-synaptic inhibitory (gephyrin), and postsynaptic excitatory (Homer1) markers (**j**, sample images; **k**, summary graphs of puncta densities plotted as averaged per experiment (top) or per region-of-interest (bottom). For additional synaptic puncta quantifications, see Fig. S3. **l, m** Nrxn3α lacking an insert in SS4 (Nrxn3α$^{SS4-}$) rescues impaired inhibitory synaptic transmission if it lacks an SS2 insert (Nrxn3α$^{SS2-}$) or contains only a short insert (Nrxn3α$^{SS2a}$) (**l**, sample traces; **m**, summary graphs of IPSC amplitudes). **n, o** Nrxn3α with an insert in SS4 (Nrxn3α$^{SS4+}$) rescues impaired inhibitory synaptic transmission only if SS2 contains no insert (Nrxn3α$^{SS2-}$) (**n**, sample traces; **o**, summary graphs of IPSC amplitudes). Numerical data are means ± SEM; n's (cells/experiments) are indicated above the sample traces (**b–i, l–o**) or in summary graph bars indicating average per independent culture on top and individual ROI's or pseudoreplicates on bottom (**k**). Statistical analyzes were performed by one-way analysis of variance (ANOVA) with Dunnett's multiple comparison test (**c, g, m**, and **o**) or two-tailed unpaired *t* test (**e, i, k**), with *$p < 0.05$, **$p < 0.01$, and ***$p < 0.001$. Source data and statistical results are provided within the Source Data file.

preferentially bind to α-neurexins containing an SS2 insert[26,29,30]. However, whether dystroglycan binding to α-neurexins is physiologically important is not known since dystroglycan binds to a large number of other ligands besides neurexins, such as agrin[31], pikachurin[32], and slit[33]. Indeed, circumstantial evidence from studies of CCK-positive synapses in the hippocampus suggested that dystroglycan-binding to neurexins is functionally insignificant[34]. Finally, a third site of alternative splicing, SS5, has also been implicated in regulating synaptic transmission, but not by a mechanism involving the regulation of ligand binding[35,36].

In brain, diverse classes of inhibitory neurons form distinct types of inhibitory synapses[37,38]. Dystroglycan is essential for the formation and/or function of a subset of these synapses[34,39–42]. Mutations in genes that are part of the dystroglycan protein complex, such as dystrophin and LARGE (the enzyme that uniquely glycosylates dystroglycan), are associated with cognitive impairments[43]. Thus, the function of dystroglycan at inhibitory synapses could account for cognitive symptoms in these patients, but the mechanism of dystroglycan's function at inhibitory synapses is enigmatic. Similarly, neurexin gene mutations have been associated with cognitive impairments in human patients. Neurexin-3 (*NRXN3*) mutations were found in families with neuropsychiatric disorders[44,45], and *NRXN3* is a class 1 gene in the SFARI autism gene database (https://gene.sfari.org/database/gene-scoring/), but again how *NRXN3* mutations predispose to cognitive impairments is unclear.

The present study was motivated by the unexpected finding that neurexin-3 (*Nrxn3*) in mice is essential for the normal release probability of inhibitory synapses in the olfactory bulb (OB), particularly at the granule cell→mitral cell (GC→MC) synapse, which constitutes one half of granule cell-mitral cell dendrodendritic synapses[35]. Here we demonstrate that only Neurexin-3α (Nrxn3α), but not Neurexin-3β (Nrxn3β), supports inhibitory synaptic transmission in dissociated olfactory bulb cultures and GC→MC synapses in vivo. We show that the function of Nrxn3α is controlled by a hierarchical splice code involving SS2 and SS4 of Nrxn3α, such that either SS2 or SS4 need to lack an insert. Unexpectedly, a minimal Nrxn3α construct that contains only a single LNS2-domain without an insert in SS2 and binds to dystroglycan fully supports inhibitory synaptic transmission in culture and GC→MC synaptic transmission in vivo; moreover, Nrxn3α performs a similar function in inhibitory synapses of the medial prefrontal cortex (mPFC) with the same dependence on LNS2-domain alternative splicing at SS2. In contrast, postsynaptic deletion of Nrxn3 in mitral cells does not impair GC→MC synaptic transmission. Both at GC→MC synapses and at inhibitory synapses of the mPFC, postsynaptic dystroglycan deletions produce a similar phenotype as presynaptic *Nrxn3* deletions. Our data thus indicate that binding of presynaptic Nrxn3α to postsynaptic dystroglycan enables a normal presynaptic release probability at inhibitory synapses, which may explain -at least in part- why mutations in *Nrxn3* and in dystroglycan-associated proteins induce cognitive impairments.

## Results

### Nrxn3α, but not Nrxn3β, restores inhibitory synapse function in *Nrxn3*-deficient OB neurons

Neurexin deletions at many synapses cause a decrease in release probability[5–7]. To explore the mechanisms involved, we focused on inhibitory synapses formed upon mitral cells in the OB, where *Nrxn3* is essential for a normal release probability[35]. We used rescue experiments to elucidate the *Nrxn3* sequences required for a normal release probability at these synapses, and tested SS4 splice variants because nearly all known functions of neurexins depend on this site of alternative splicing.

We infected mixed neuron-glia cultures from the OB of *Nrxn3* cKO mice with lentiviruses expressing inactive (ΔCre, as a control) or active Cre-recombinase (Cre), without or with co-expression of Nrxn3α$^{SS4+}$ or Nrxn3α$^{SS4-}$ (containing or lacking an insert in SS4, respectively), or Nrxn3β$^{SS4+}$ (containing an insert in SS4) (Fig. 1a). To permit neuron-specific expression, we used the human synapsin-1 promoter. We then performed whole-cell patch-clamp recordings from larger mitral/tufted cells (referred to as MCs), which can be clearly distinguished from smaller inhibitory granule cells and other neurons[35,46,47], and used extracellular stimulation to evoke IPSCs. Since granule cells (GCs) are by far the most abundant inhibitory neurons in the OB (comprising ~82%, see Fig. S3e)[48], the evoked IPSCs largely reflect GC→MC synaptic transmission in cultured OB neurons, although they likely also contain contributions from other inhibitory neurons.

The *Nrxn3* deletion severely impaired (60–80% decrease) evoked IPSCs in cultured OB neurons (Fig. 1b, c), consistent with earlier results[35]. This impairment was rescued by Nrxn3α$^{SS4+}$ or Nrxn3α$^{SS4-}$, which did not affect evoked IPSCs in control neurons (Figs. 1b, c, S1a). Nrxn3β$^{SS4+}$, however, suppressed evoked IPSCs both in *Nrxn3*-deficient and in control neurons (Fig. 1b, c). In the initial experiments, this effect was not statistically significant, but independent replication experiments confirmed that expression of Nrxn3β$^{SS4+}$ in WT neurons decreased evoked IPSC amplitudes ~50% (Figs. 1d, e, S1c). Measurements of evoked excitatory synaptic transmission, monitored as NMDAR-dependent evoked EPSCs, failed to detect any changes induced by the *Nrxn3* deletion or by expression of Nrxn3α$^{SS4+}$, Nrxn3α$^{SS4-}$, or Nrxn3β$^{SS4+}$, suggesting that the effect of the *Nrxn3* deletion is specific for inhibitory synapses in OB neurons (Figs. 1f–i, S1b, d).

In agreement with the results on evoked IPSCs, deletion of *Nrxn3* greatly lowered (50–80% decrease depending on the experiment) the spontaneous mIPSC frequency but not the mIPSC amplitude in mitral/tufted cells in OB cultures (Fig. S1e-S1i). This decrease was also rescued by expression of Nrxn3α$^{SS4+}$ or Nrxn3α$^{SS4-}$ (Fig. S1h, i), whereas expression of Nrxn3β$^{SS4+}$ decreased the mIPSC frequency again both in *Nrxn3*-deficient and control neurons (Fig. S1h–k). The impairment in inhibitory neuron→MC synaptic transmission in cultured *Nrxn3*-deficient OB neurons was not due to a decrease in synapse numbers or GABA$_A$-receptor function because no change in inhibitory synapse

numbers or synaptic surface GABA$_A$-receptor levels was detected (Figs. 1j, k, S2). This includes inhibitory synapses co-labeled without or with Homer1 puncta, corresponding to putative non-reciprocal and reciprocal synapses, respectively.

Viewed together, these data suggest that in olfactory inhibitory synapses, *Nrxn3* is essential for synaptic transmission in a manner that can be rescued by Nrxn3α$^{SS4+}$ or Nrxn3α$^{SS4-}$, but not by Nrxn3β$^{SS4+}$. Across the brain, neurexins exhibit dynamic regulation of SS4 alternative splicing, with the OB and cerebellum expressing almost exclusively SS4 + variants (Fig. S3a–d). RT-PCR measurements revealed that Nrxn3α$^{SS4+}$ is highly enriched in inhibitory OB neurons, ~82% of which are granule cells[48] (Fig. S3e), as measured using RiboTag pulldowns of translating mRNAs in vGAT-positive neurons (Fig. S3). In contrast, Nrxn3β$^{SS4+}$ and Nrxn3β$^{SS4-}$ are nearly undetectable, suggesting that the dominant-negative action of Nrxn3β$^{SS4+}$ is likely not physiologically important for the OB (Fig. S3).

### A combinatorial splice controls Nrxn3α function at inhibitory synapses

Since Nrxn3α rescued the *Nrxn3* KO phenotype independent of SS4 alternative splicing, we turned our attention to the only other alternatively spliced sequence of neurexins that is well established to regulate ligand interactions, SS2. SS2 is present in the LNS2 domain of α-neurexins, which is lacking in β-neurexins[21], and controls binding of neurexophilins and dystroglycan to neurexins[26–30,49–51]. SS2 is expressed in three variants, SS2- lacking an insert, SS2a containing an 8 amino-acid 'a' insert, and SS2ab containing an additional 7 amino-acid 'b' insert[21].

RT-PCR measurements revealed that in most brain regions, *Nrxn1* and *Nrxn3* are predominantly expressed as SS2- variants, whereas approximately half of the *Nrxn2* mRNAs are present as SS2a variants (Fig. S4a–d). SS2ab variants are uniformly rare (<10%) for all neurexins and all brain regions. In the OB, Nrxn3α is predominantly expressed as the Nrxn3α$^{SS2-}$ variant, with Nrxn3α$^{SS2a}$ accounting for ~10% and Nrxn3α$^{SS2ab}$ < 4% of transcripts (Fig. S4a–d). Using RT-PCR with mRNAs isolated by RiboTag purification from inhibitory neurons (~83% of which are granule cells, Fig. S3) and from mitral cells of the OB, we found that Nrxn3α is expressed in inhibitory neurons exclusively as the Nrxn3α$^{SS2-}$ variant (>99%), whereas in mitral cells Nrxn3α$^{SS2-}$ and Nrxn3α$^{SS2a}$ variants are produced almost equally (~55% vs 45%; Fig. S4e, S4f). Thus, SS2 alternative splicing of Nrxn3 is highly regulated in the OB, but its pattern of alternative splicing is distinct from that of SS4 which is present as the SS4 + variant in the entire OB (Fig. S3).

Does alternative splicing at SS2 regulate the ability of Nrxn3α to rescue the *Nrxn3* KO phenotype in inhibitory synapses? To address this question, we probed the effects of both SS1 and SS2 alternative splicing on inhibitory neuron→MC synaptic transmission in cultured neurons. We included SS1 alternative splicing in the analysis because SS1 is dynamically spliced in the brain[18,19,52] and its adjacency to SS2 may have a peripheral effect on neurexophilin-binding to neurexins[29].

Rescue experiments of *Nrxn3* KO neurons from the OB with different SS1, SS2, and SS4 variants of Nrxn3α uncovered a surprising finding: When SS4 lacked an insert, both Nrxn3α$^{SS2-}$ and Nrxn3α$^{SS2a}$ rescued, but Nrxn3α$^{SS2ab}$ was inactive (Figs. 1l, m, S4g). However, when SS4 contained an insert as in almost all *Nrxn3* transcripts in the OB, only Nrxn3α$^{SS2-}$ but not Nrxn3α$^{SS2a}$ or Nrxn3α$^{SS2ab}$ was able to rescue (Figs. 1n, o, S4h). SS1 alternative splicing had no effect on rescue. Since *Nrxn3* is expressed in inhibitory neurons, including granule cells, exclusively as the SS4 + variant (Fig. S3h), *Nrxn3* alternative splicing at SS2 controls *Nrxn3* function at inhibitory neuron→MC synapses, at least as monitored in cultured neurons. Overall, these data suggest that alternative splicing of Nrxn3α at SS2 and SS4 govern its function at inhibitory neuron→MC synapses of the OB, such that either SS2 or SS4 (or both) needs to lack an insert in order for Nrxn3α to sustain synaptic function.

### Nrxn3α-LNS2$^{SS2-}$ containing a single LNS-domain fully sustains inhibitory synapse function

We constructed a series of deletion constructs of Nrxn3α to determine which domains are required to support inhibitory neuron→MC synaptic transmission (Fig. 2a, d). Rescue experiments with constructs that include various extracellular LNS- and EGF-domains revealed that a minimal Nrxn3α protein containing only a single LNS domain, the LNS2 domain in which SS2 is located, was sufficient to reverse the *Nrxn3α* KO phenotype (Figs. 2b–h, S4i–k). In the minimal Nrxn3α-LNS2 constructs, the LNS2 domain is fused to the glycosylated stalk region of Nrxn3α that separates the last LNS domain from the membrane and that is followed by the transmembrane region and cytoplasmic tail (Fig. 2d). Importantly, the minimal Nrxn3α-LNS2 construct only rescued inhibitory synaptic transmission in *Nrxn3* KO neurons when SS2 in the LNS2-domain lacked an insert (Nrxn3α-LNS2$^{SS2-}$). Even the introduction of the short 'a' insert (Nrxn3α-LNS2$^{SS2a}$) completely abolished rescue (Figs. 2g, h, S4k). Thus, surprisingly, a single LNS domain of Nrxn3α is sufficient to maintain inhibitory synaptic transmission, indicating that most domains comprising the Nrxn3α architecture are dispensable for its function within inhibitory neuron→MC synapses.

### Nrxn3α-LNS2$^{SS2-}$ sustains GC→MC synaptic transmission in vivo without affecting synapse numbers

The finding that Nrxn3α-LNS2$^{SS2-}$ is sufficient to rescue inhibitory neuron→MC synaptic transmission in cultured OB neurons is surprising, raising the possibility of experimental artifacts. Moreover, although granule cells are the most prevalent type of inhibitory neuron in the OB, culture experiments do not permit us to specifically test GC→MC synaptic transmission. Thus, we decided to examine the phenotype of the *Nrxn3* deletion and its rescue by Nrxn3α-LNS2$^{SS2-}$ in GC→MC synapses in vivo.

We stereotactically infected the OB of *Nrxn3* cKO mice at P21 with AAVs encoding either ΔCre (as a control) or Cre (to delete *Nrxn3*), without or with Nrxn3α-LNS2$^{SS2-}$ or Nrxn3α-LNS2$^{SS2a}$ rescue constructs (Fig. 3a). ΔCre and Cre were expressed as tdTomato fusion proteins to visualize AAV-infected cells. Morphological studies of OB sections of infected mice at P35-42 showed that tdTomato was nearly ubiquitously expressed (Fig. 3b). The Nrxn3α-LNS2$^{SS2-}$ and Nrxn3α-LNS2$^{SS2a}$ constructs include an extracellular N-terminal HA-epitope tag, enabling us to visualize them by immunocytochemistry of OB sections (Fig. 3c). The two minimal Nrxn3α-LNS2 proteins were similarly distributed in the synaptic strata of the OB (Fig. 3c), suggesting that this approach is well suited for probing *Nrxn3* function in the OB in vivo.

We next examined the effect of the *Nrxn3* KO and of the Nrxn3α-LNS2$^{SS2-}$ and Nrxn3α-LNS2$^{SS2a}$ rescue in the OB on the density of GC→MC synapses. We stained OB sections for synaptoporin and gephyrin, which are specific markers for GC→MC synapses in particular, and for inhibitory synapses in general[53–55]. We then analyzed the density of gephyrin- and synaptoporin-positive puncta as well as that of puncta containing both signals (Fig. 3d, S5a). The results were analyzed using either the number of mice or the number of regions-of-interest (ROI's) as the statistical 'n' because the former is likely more correct, but the latter is a common practice despite the fact that the number of ROI's not actually true replicates, but represent pseudo-replicates that boost statistical power. Using both statistical approaches, we detected no change in GC→MC synapse density except for a statistically significant 10% decrease using pseudo-replicate quantifications that was observed only for the gephyrin puncta density measurements (Figs. 3e, S5b). Viewed together, these results indicate that the *Nrxn3* KO in the OB does not cause a major change in synapse numbers.

Do in vivo deletions of *Nrxn3* cause an impairment in GC→MC synapse function that can be rescued by the minimal Nrxn3α-LNS2$^{SS2-}$ construct, analogous to what we observed with inhibitory neuron→MC synapses in OB cultures? We addressed this critical question using whole-cell patch-clamp recordings from mitral cells in acute OB slices.

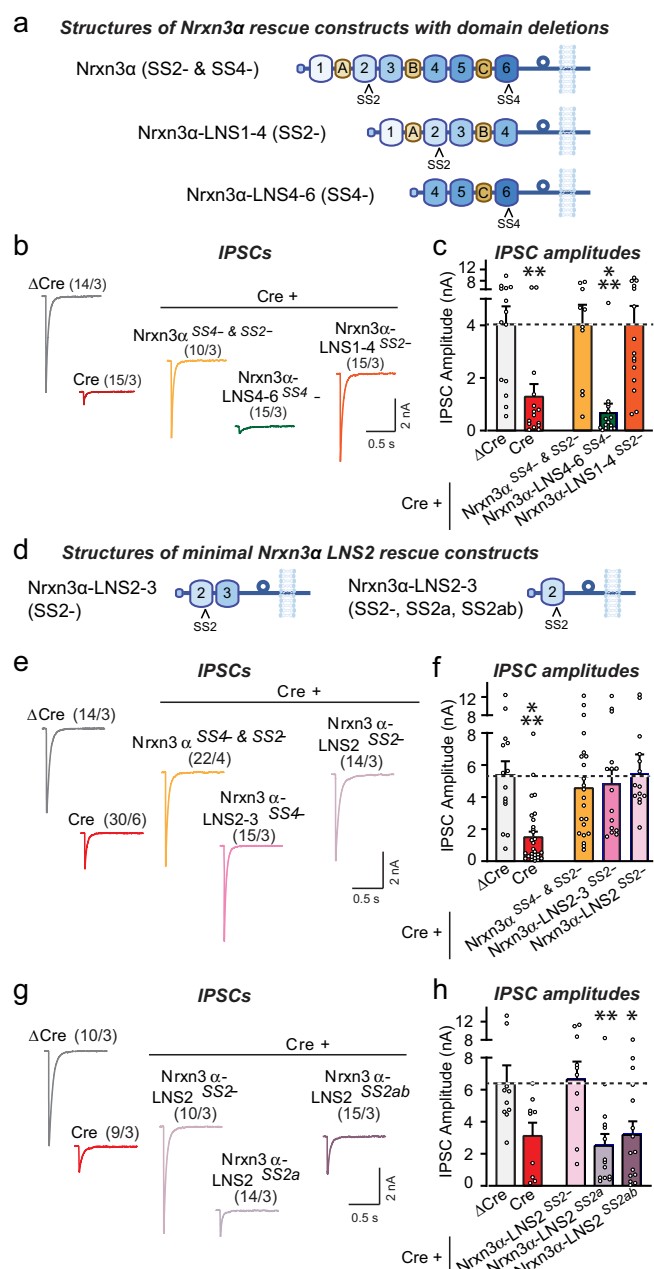

**Fig. 2 | A minimal protein composed of the single LNS2-domain of Nrxn3α attached to its C-terminal stalk sequence, transmembrane region and cytoplasmic sequence fully rescues inhibitory synaptic transmission in cultured Nrxn3-deficient OB neurons.** All experiments were performed in dissociated OB cultures obtained from newborn *Nrxn3* cKO mice that were infected with lentiviruses expressing ΔCre (control) or Cre without or with the indicated rescue proteins. **a** Schematic of Nrxn3α rescue constructs with domain deletions. **b**, **c** Nrxn3α lacking LNS5 and LNS6 domains fully rescues impaired inhibitory synaptic transmission in *Nrxn3*-deficient OB neurons, whereas Nrxn3α lacking LNS1-3 domains does not (b, sample traces; c, summary graphs of IPSC amplitudes). **d** Schematic of minimal Nrxn3α rescue constructs. **e**, **f** A minimal Nrxn3α protein containing only LNS2 without an SS2 insert linked to the C-terminal Nrxn3α sequences, rescues impaired inhibitory synaptic transmission in *Nrxn3*-deficient OB neurons (e, sample traces; f, summary graphs of IPSC amplitudes). **g**, **h** The minimal Nrxn3α protein containing only LNS2 is unable to rescue inhibitory synaptic transmission in *Nrxn3*-deficient OB neurons if SS2 contains an insert (g, sample traces; h, summary graphs of IPSC amplitudes). Numerical data are means ± SEM; n's (cells/experiments) are indicated above the sample traces and apply to all graphs in an experimental series. Statistical analyzes were performed with a one-way analysis of variance (ANOVA) with Dunnett's multiple comparison test, with *$p < 0.05$, **$p < 0.01$, and ***$p < 0.001$. Source data and statistical results for all experiments are provided within the Source Data file.

To selectively probe GC→MC synaptic transmission, we measured evoked IPSCs elicited by extracellular simulation of the granule cell layer in acute slices. This stimulation paradigm allowed us to avoid activation of other interneurons that synapse upon mitral cells, including periglomerular and EPL interneurons. To control for possible effects caused by variations in the placement of the stimulating electrode, we analyzed evoked IPSCs as input/output curves (Fig. 4d). Again, the *Nrxn3* KO caused a large impairment (~50% decrease) that was fully reversed by Nrxn3α-LNS2^SS2- but not by Nrxn3α-LNS2^SS2ab (Fig. 4d–f). No change in IPSC kinetics was detectable (Fig. 4g). However, we found a large increase in the coefficient of variation of IPSCs that also was rescued by Nrxn3α-LNS2^SS2- but not by Nrxn3α-LNS2^SS2ab (Fig. 4h). This increase is indicative of a decrease in release probability[58]. To confirm a decrease in release probability, we measured the paired-pulse ratio of GC→MC IPSCs as a function of the interstimulus interval (Fig. 4i). The *Nrxn3* KO caused a massive increase in the paired-pulse ratio consistent with a decrease in release probability; this increase was also completely reversed by Nrxn3α-LNS2^SS2- but not by Nrxn3α-LNS2^SS2ab (Fig. 4j). Although neurexins are well established to act presynaptically, we tested a possible postsynaptic contribution of Nrxn3 at the GC→MC synapse by investigating the effect of selective postsynaptic deletion of *Nrxn3* in mitral cells. For this purpose, we infected the piriform cortex of *Nrxn3* cKO mice with retro-AAVs encoding ΔCre or Cre fused to tdTomato[59]. The retro-AAVs infect the mitral cell axon terminals in the piriform cortex, thereby inducing selective expression of ΔCre or Cre in mitral cells of the OB that can be identified by the presence of tdTomato. We detected no change in GC→MC synaptic transmission after postsynaptic deletion of *Nrxn3* in mitral cells (Fig. S6), indicating that Nrxn3 acts presynaptically in regulating inhibitory synaptic transmission. Viewed together, these data show that the in vivo deletion of *Nrxn3* severely impairs GC→MC synaptic transmission by suppressing, at least in part, the release probability. Moreover, consistent with our findings in cultured neurons, these data show that a minimal Nrxn3α construct containing only the LNS2 domain can rescue this impairment in a manner regulated by alternative splicing at SS2.

### Impaired inhibitory synaptic strength in *Nrxn3*-deficient mPFC neurons is also rescued by the minimal Nrxn3α-LNS2^SS2- construct

GC→MC synapses of the OB are part of reciprocal dendrodendritic synapses that may differ from 'standard' inhibitory synapses in the

In addition to receiving extensive synaptic inhibition from granule cells, MCs also receive synaptic inhibition from periglomerular cells and EPL interneurons[56]. Thus, mIPSCs monitored in MCs include events produced by non-GC inhibitory neurons, though these synapses are much fewer in numbers and are located further from the soma, suggesting that their responses will be poorly detected due to dendritic filtering. No change in the passive electrical properties of mitral cells was evident after the *Nrxn3* KO (Fig. S5c, d). Recordings of mIPSCs showed that the in vivo *Nrxn3* KO robustly decreased (60%) the mIPSC frequency; this decrease was fully rescued by Nrxn3α-LNS2^SS2- but not by Nrxn3α-LNS2^SS2ab (Fig. 4a, b). No significant change in the mIPSC amplitude or kinetics was detected (Figs. 4c, S5e). A recent study found that Nrxn3 differentially regulates the formation and/or function of a subset of inhibitory synapses in the ventral subiculum of the hippocampus in a sex-dependent manner[57]. However, separate analyzes of slices from male and female mice exhibited a similar decrease in mIPSC frequency (Fig. S5f), indicating that not all contributions of Nrxn3 to inhibitory synapse function are sex-dependent.

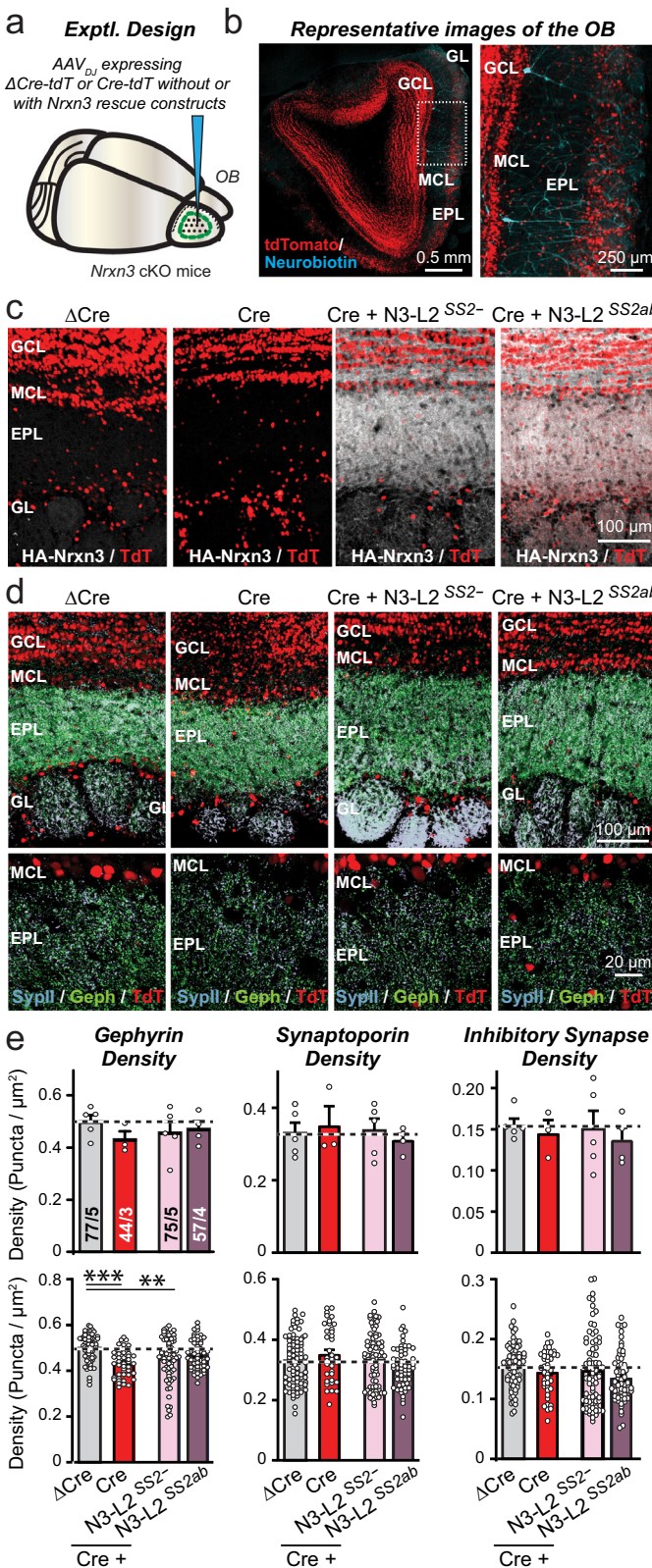

**a** *Exptl. Design*

AAV$_{DJ}$ expressing
ΔCre-tdT or Cre-tdT without or
with Nrxn3 rescue constructs

OB

*Nrxn3 cKO mice*

**b** *Representative images of the OB*

GL / GCL / MCL / EPL

tdTomato / Neurobiotin 0.5 mm 250 µm

**c** ΔCre | Cre | Cre + N3-L2$^{SS2-}$ | Cre + N3-L2$^{SS2ab}$

GCL / MCL / EPL / GL

HA-Nrxn3 / TdT 100 µm

**d** ΔCre | Cre | Cre + N3-L2$^{SS2-}$ | Cre + N3-L2$^{SS2ab}$

GCL / MCL / EPL / GL

MCL / EPL

SypII / Geph / TdT 100 µm 20 µm

**e** *Gephyrin Density* | *Synaptoporin Density* | *Inhibitory Synapse Density*

(top row) Density (Puncta / µm²): Gephyrin axis to 0.6 (77/5, 44/3, 75/5, 57/4); Synaptoporin axis to 0.4; Inhibitory Synapse axis to 0.20

(bottom row) Density (Puncta / µm²): Gephyrin axis to 0.8 (*** **); Synaptoporin axis to 0.6; Inhibitory Synapse axis to 0.3

x-axis categories: ΔCre, Cre, N3-L2$^{SS2-}$, N3-L2$^{SS2ab}$ (Cre +)

**Fig. 3 | Conditional in vivo deletion of Nrxn3 in the OB has no effect on synapse density, nor does rescue of the Nrxn3 deletion by minimal Nrxn3 LNS-domain constructs. a, b** Experimental design of in vivo *Nrxn3* deletions and rescues. The OB was stereotactically infected with AAVs expressing ΔCre (control), Cre, or Cre with additional AAVs encoding Nrxn3-LNS2 rescue constructs (**a**, schematic of stereotactic injections; **b**, representative fluorescence image of an OB section that was infected with AAVs expressing tdTomato fused to Cre, with subsequent patching of mitral cells that were filled with neurobiotin (blue)). Neuron-specific expression of ΔCre/Cre and rescue constructs was achieved using the synapsin-1 promoter. Note that AAVs infect granule cells more efficiently than mitral cells (see panel b). **c** Minimal Nrxn3α-LNS2 rescue proteins are localized to synaptic layers in the OB after expression via AAVs, as visualized by immunocytochemistry for the N-terminal HA-epitope contained in the constructs (white, HA-Nrxn3α-LNS2 proteins; red, tdTomato). **d, e** Conditional *Nrxn3* deletion and rescue with minimal Nrxn3α-LNS2 constructs does not alter inhibitory synapse numbers in vivo. Sections from mice (infected as shown in A) were analyzed by quantitative immunocytochemistry for the presynaptic marker synaptoporin (a.k.a. synaptophysin-2; light blue) that is specific for granule cell→mitral cell synapses in the OB, and for the postsynaptic inhibitory synapse marker gephyrin (green) (**d**, sample images; **e**, summary graphs of puncta densities). Data are means ± SEM; n's (cells/experiments) indicated in summary graph bars apply to all graphs in an experimental series. Statistical analyzes using one-way ANOVA with Tukey's multiple comparison test (**e**). Appropriate HA labeling in c was confirmed in tissue from all animals quantified in e. Puncta densities in e are analyzed both per animal (top) and per region-of-interest (bottom); statistical significance is observed for gephyrin staining but not the other parameters when regions-of-interest are used as n's because pseudo-replicates in this analysis boost statistical significance independent of the actual number of experiments. Source data and statistical results for all experiments are provided within the Source Data file.

amplitude and synaptic charge transfer of evoked IPSCs in these neurons (Fig. S7a, b). This decrease was rescued both by full-length Nrxn3α lacking inserts in SS2 and SS4, and by the minimal Nrxn3α-LNS2$^{SS2-}$ protein (Fig. S7a, b).

Next, we deleted *Nrxn3* from mPFC neurons in vivo using stereotactic injections of AAVs expressing ΔCre (as a control) or Cre, with or without co-expression of Nrxn3α-LNS2$^{SS2-}$ similar to the in vivo OB experiments (Fig. 5a). A total of 2–3 weeks after infection, we sectioned acute slices from the mice and patched Layer 5/6 neurons for electrophysiological recordings (Fig. 5b). Measurements of spontaneous mIPSCs uncovered a robust decrease in mIPSC frequency (-25%) in *Nrxn3*-deficient neurons (Fig. 5c, d), suggesting that a subset of the heterogeneous inhibitory synaptic inputs on Layer 5/6 neurons may have been impaired by the *Nrxn3* KO similar to GC→MC synapses. Expression of Nrxn3α-LNS2$^{SS2-}$ fully rescued the decrease in mIPSC frequency in *Nrxn3* KO synapses. Moreover, we observed a small decrease in the mIPSC amplitude induced by the *Nrxn3* KO that was not rescued by the Nrxn3α-LNS2$^{SS2-}$ construct (Fig. 5e), suggesting a different additional role for *Nrxn3* in postsynaptic GABA$_A$R function in the mPFC. Such a role was not detected in our in vivo OB experiments, consistent with the notion that neurexins and their ligands are expressed in distinct combinatorial patterns in various types of neurons and thus different degrees of redundancy may occur among neurexins and their ligands in these types of neurons. This notion is further supported by our previous observation that a significant loss of GABA$_A$R function was observed following the deletion of neuroligins in mitral cells of the OB[59]. No changes in the passive electrical properties of the pyramidal mPFC neurons or in the mIPSC kinetics were detected (Fig. S7c–e).

Next, we monitored evoked IPSCs, again using input/output measurements to control for possible variations in the placement of the stimulating electrode, even though -as always- all experiments were conducted 'blindly' (Fig. 5f). The *Nrxn3* deletion greatly suppressed the synaptic strength of evoked IPSCs (-50% decrease), which could be fully rescued by the Nrxn3α-LNS2$^{SS2-}$ construct (Fig. 5g, h). In addition, the *Nrxn3* KO caused a slowing of the IPSC rise but not decay

CNS. The surprising finding that alternative splicing of Nrxn3α regulates this synapse via the activity of a single LNS domain could represent an exceptional mechanism that is specific to dendrodendritic synapses and not shared by other inhibitory synapses.

To ask whether the LNS2-dependent function of *Nrxn3* at GC→MC synapses of the OB may apply to other types of inhibitory synapses, we first tested cultured cortical neurons. The *Nrxn3* KO nearly halved the

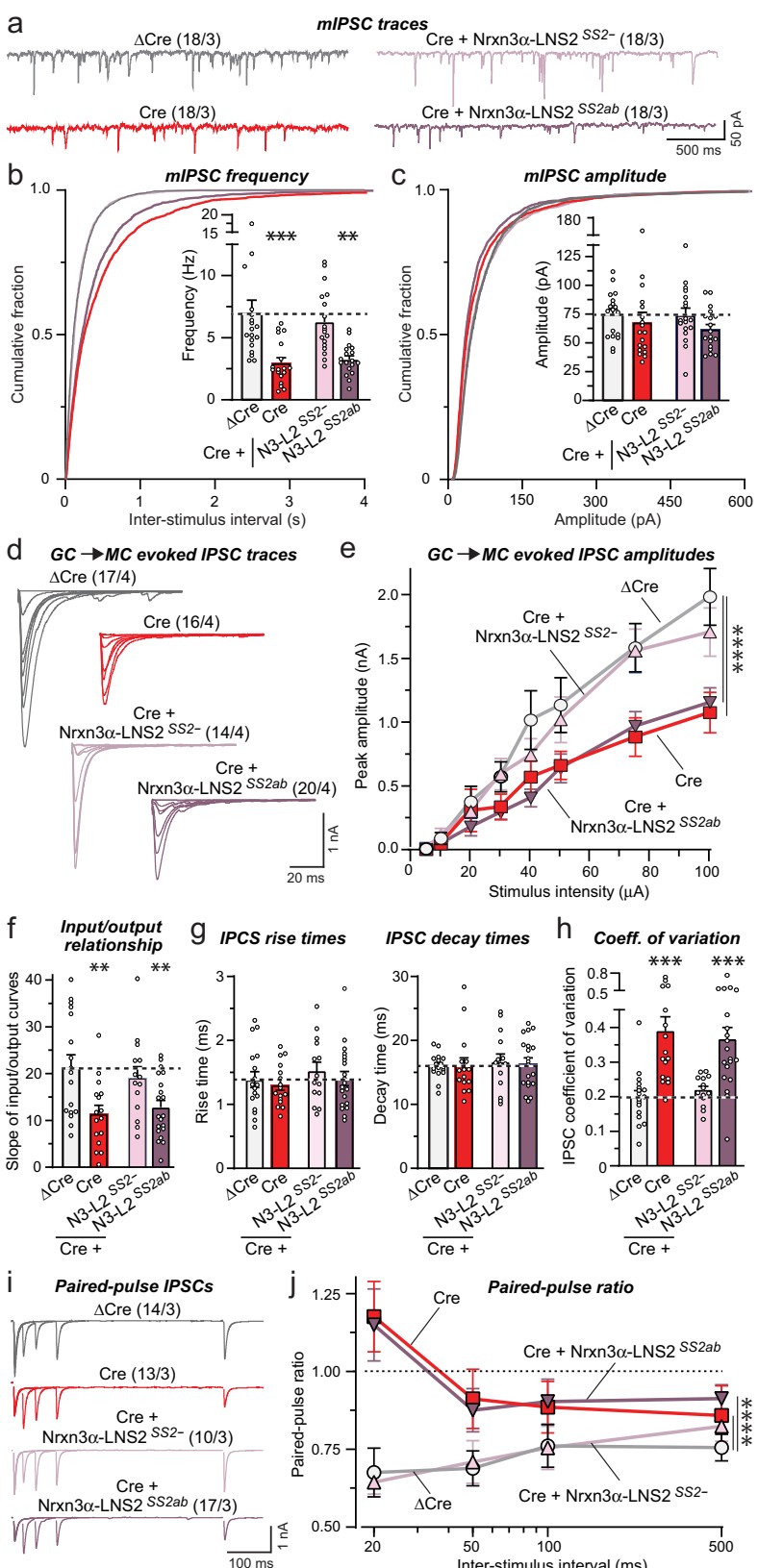

times, again with full rescue by Nrxn3α-LNS2^SS2- (Fig. 5i). Finally, the *Nrxn3* KO induced a large increase (-120%) in the coefficient of variation of evoked IPSCs in mPFC synapses similar to OB GC→MC synapses, and this phenotype was also reversed by Nrxn3α-LNS2^SS2- (Fig. 5j). Together these data indicate that Nrxn3α performs a similar function

in a subset of inhibitory synapses in the mPFC as in GC→MC synapses in the OB, namely an essential role in sustaining the normal release probability at these synapses, such that the *Nrxn3* deletion ablates nearly half of the inhibitory synaptic strength in a manner that can be rescued by the Nrxn3α-LNS2^SS2- construct.

**Fig. 4 | Conditional in vivo deletion of Nrxn3 in the OB severely impairs GC→MC inhibitory synaptic transmission by lowering the release probability: Rescue by minimal Nrxn3-LNS2 constructs lacking an insert in SS2.** All experiments were performed by patch-clamp recordings from mitral cells in acute slices from *Nrxn3* cKO mice whose OB was infected with AAVs (see Fig. 3a, S5c–e). **a–c** The *Nrxn3* deletion decreases the mIPSC frequency; this decrease is rescued only by the minimal Nrxn3α-LNS2 construct lacking an insert in SS2 (**a**, representative mIPSC traces recorded in the presence of TTX; **b, c** cumulative probability of the mIPSC interevent intervals and amplitudes, insets: summary of the mIPSC frequency and amplitudes). **d–f** The *Nrxn3* deletion decreases the evoked IPSC amplitude as documented by input/output curves. This decrease is rescued only by the minimal Nrxn3α-LNS2 construct without an insert in SS2 (**d**, representative IPSC traces; e, summary of input/output amplitudes; **f**, summary of the slope of input/output curves). **g** The *Nrxn3* deletion and expression of rescue constructs have no effect on

evoked IPSC kinetics (summary of the IPSC rise (left) and decay times (right)). **h** The *Nrxn3* deletion increased the coefficient of variation of IPSCs, suggesting a decrease in release probability; this phenotype is rescued only by the minimal Nrxn3α-LNS2 construct without an insert in SS2. **i, j** The *Nrxn3* deletion induces a large increase in the paired-pulse ratio; this phenotype is rescued by the minimal Nrxn3α-LNS2 construct without an insert in SS2 (**i**, representative traces; **j**, summary of the paired-pulse ratio). Numerical data are means ± SEM; n's (cells/experiments) are indicated above the sample traces and apply to all graphs in an experimental series. Statistical analyzes were performed using two-way ANOVA in **e** and **j** and one-way ANOVA in **b, c**, and **f–h** with Dunnett's and Tukey's multiple comparison test respectively with regards to the ΔCre group, with *$p < 0.05$, **$p < 0.01$, ***$p < 0.001$, ***$p < 0.001$, and ****$p < 0.0001$. Source data and statistical results for all experiments are provided within the Source Data file.

## CRISPRi-mediated inhibition of dystroglycan expression phenocopies the *Nrxn3* KO at GC→MC synapses in cultured OB neurons

The requirement and sufficiency of the minimal Nrxn3α-LNS2$^{SS2-}$ construct that contains a single extracellular interaction domain (LNS2) with a specific splice variant (SS2-) for synaptic transmission at a subset of inhibitory synapses suggests that Nrxn3α functions by binding to a trans-synaptic ligand. At present, only one ligand is known to specifically bind to the LNS2 domain of neurexins lacking an insert in SS2: dystroglycan[26,30]. Neurexophilin also binds to the LNS2 domain, but its binding is enhanced instead of impeded by an insert in SS2[27–29]. Notably, dystroglycan also binds to the LNS6 domain of α-neurexins when the LNS6 domain lacks an insert in SS4, accounting for the finding that full-length Nrxn3α is still functional at inhibitory neuron→MC synapses in OB cultures when it contains a partial insert in SS2 (i.e., SS2a[21]), as long as SS4 lacks an insert (Fig. 1)[26].

To explore the possibility that dystroglycan may be the postsynaptic ligand for presynaptic Nrxn3α at GC→MC synapses that is required for sustaining their release probability, we selected a guide RNA (gRNA) that enables potent CRISPR interference (CRISPRi)-mediated inhibition of dystroglycan expression in cultured OB neurons (Fig. 6a). Although an apparently incomplete suppression of dystroglycan mRNA level (~65% decrease) was observed, this is likely an underestimate of the neuronal dystroglycan suppression since the lentiviral expression of the CRISPRi components is most efficient in neurons. Electrophysiological recordings revealed that the CRISPRi-mediated partial inhibition of dystroglycan expression induced a robust decrease (~60%) in the amplitude of evoked IPSCs (Fig. 6b, c).

The suppression of IPSCs by the inhibition of dystroglycan expression in cultured OB neurons (Fig. 6b, c) is similar to that observed for the *Nrxn3* KO (Fig. 1b, c). To test whether these two adhesion molecules operate in the same pathway, we compared the phenotypes of single and double dystroglycan and *Nrxn3* KOs by combining the conditional deletion of *Nrxn3* with the CRISPRi-mediated inhibition of dystroglycan expression in cultured OB neurons from *Nrxn3* cKO mice. We infected the neurons with lentiviruses expressing either ΔCre (control) or Cre (to delete *Nrxn3*) and/or dCAS9-KRAB and the dystroglycan gRNA, and measured evoked IPSCs and NMDAR- and AMPAR-mediated EPSCs. The dystroglycan and *Nrxn3* deletions individually and together induced a 60-70% decrease in IPSC amplitudes and charge transfer, with no aggravation of the phenotype by the combined deletion compared to the individual deletions (Fig. 6d–f). None of the deletions, individually or combined, had a significant effect on NMDAR- or AMPAR-EPSCs (Fig. 6g–i). These data suggest that *Nrxn3* and dystroglycan act in the same pathway to sustain inhibitory neuron→MC synaptic transmission, consistent with the notion that they function by binding to each other.

## Synaptic localization of dystroglycan in the OB

In the OB, dystroglycan is prominently expressed by mitral cells where it was localized to GC→MC reciprocal synapses via immunoelectron microscopy[60]. Consistent with the hypothesis that dystroglycan is the postsynaptic receptor for Nrxn3 in mitral cells, *Nrxn3* mRNA is relatively abundant in the granule cell layer, while dystroglycan (*Dag1*) mRNA is enriched in mitral cells (Fig. S8a, b)[61]. To confirm the synaptic localization of dystroglycan in the OB, we optimized staining for dystroglycan with the IIH6C4 monoclonal antibody using different times of post-fixation with 4% PFA (i.e., overnight, 20 minutes, and 10 minutes). We found that only light post-fixation (i.e., less than 20 minutes) permitted robust detection of dystroglycan in the synaptic neuropil and around blood vessels (Figs. 7, S8c). It is well known that excessive cross-linking can hinder access of epitopes needed to localize proteins located within the synaptic cleft[62]. Using stimulated emission depletion (STED) super-resolution microscopy, we found that dystroglycan nanoclusters were abundant at inhibitory synapses in the EPL of the OB, including reciprocal synapses, and in large inhibitory synapses located within glomeruli (Figs. 7b–d, S8d–g). Given that STED was only performed in 2D and synapses were viewed at random angles, the actual number of inhibitory synapses with dystroglycan nanoclusters is likely much higher. Consistent with prior studies[34,41,42], using light fixation conditions we also found that IIH6C4 labeled dystroglycan at a subset of inhibitory synapses in the hippocampus and cerebellum (Figs. 7e–h, S8h–i). Thus, the localization of dystroglycan in the OB is consistent with a potential role as a postsynaptic ligand of Nrxn3 at GC→MC synapses.

## Postsynaptic dystroglycan deletion in vivo recapitulates the *Nrxn3* KO phenotype in the OB and mPFC

To validate the results obtained with the inhibition of dystroglycan expression in cultured OB neurons, we next investigated the effect of a CRISPR-mediated deletion of dystroglycan in vivo in both the OB and the mPFC (Figs. 8, 9). We used CRISPR-mediated deletions instead of CRISPRi in these experiments because the components needed for CRISPRi could not be encoded by a single AAV. In the first set of experiments, we infected the OB of CAS9-expressing mice with AAVs encoding the dystroglycan gRNA or a control gRNA and tdTomato, and examined the efficiency of the dystroglycan deletion and the effect of the deletion on the inhibitory synapse density (Figs. 8a–d, S9a–i). As assessed by immunocytochemistry for dystroglycan, the CRISPR-mediated dystroglycan deletion was efficient with a ~60% decline in total dystroglycan signal (Fig. S9a, b). Again, this is likely an underestimate of the degree of the deletion of dystroglycans since our AAVs are optimized for neuronal expression but much of the dystroglycan in brain is expressed in cells surrounding blood vessels. Unlike localization experiments with the well-validated IIH6C4 monoclonal antibody (Figs. 7, S8), for this experiment, we employed a rabbit monoclonal antibody against α-Dystroglycan[63] to avoid unspecific mouse

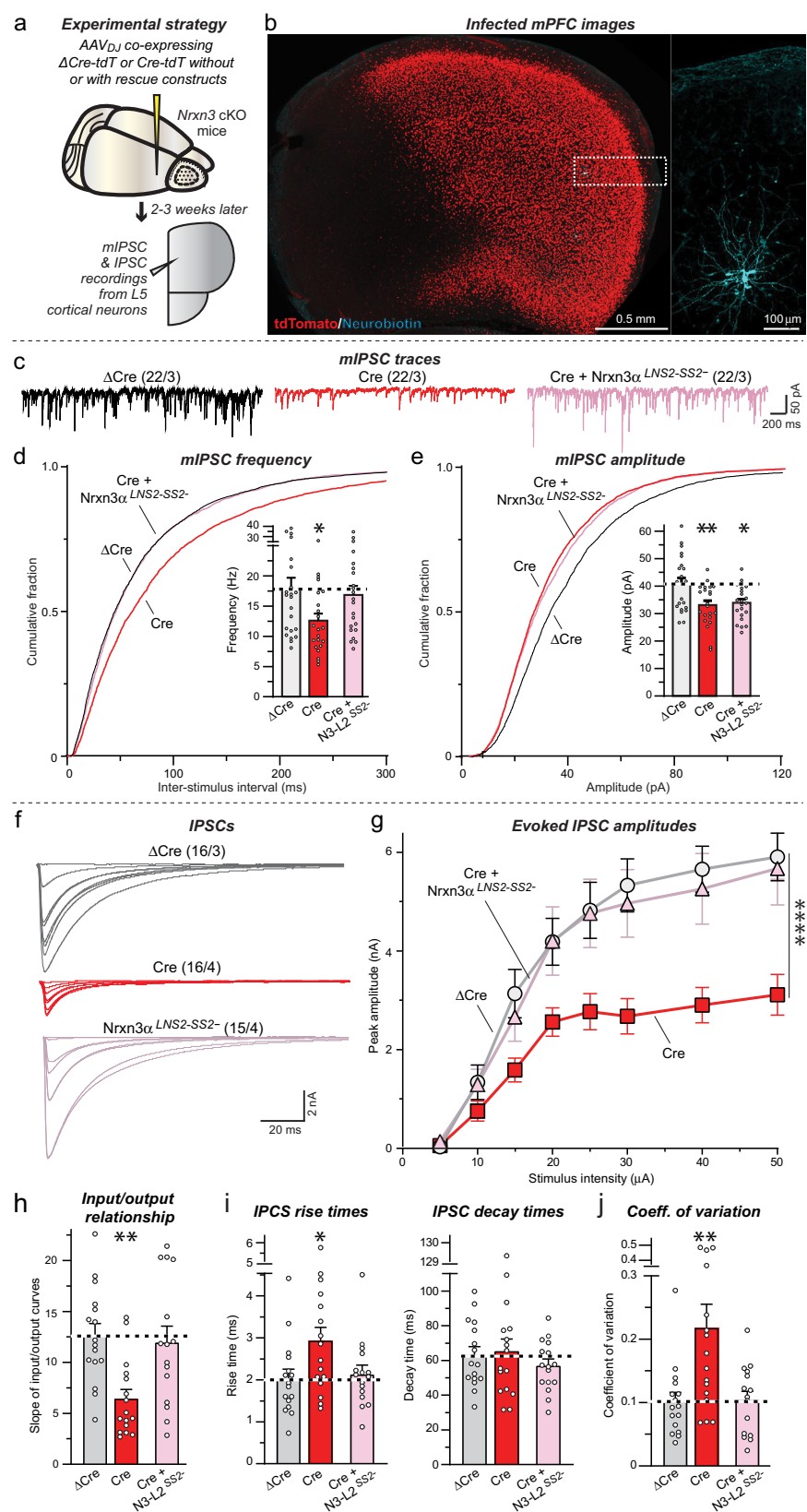

**a** *Experimental strategy*
*AAV_DJ co-expressing ΔCre-tdT or Cre-tdT without or with rescue constructs*

*Nrxn3 cKO mice*

2-3 weeks later

*mIPSC & IPSC recordings from L5 cortical neurons*

**b** *Infected mPFC images*

tdTomato/Neurobiotin    0.5 mm    100 µm

**c** *mIPSC traces*

ΔCre (22/3)    Cre (22/3)    Cre + Nrxn3α^LNS2-SS2- (22/3)

50 pA / 200 ms

**d** *mIPSC frequency*

Cumulative fraction vs Inter-stimulus interval (ms)

Frequency (Hz): ΔCre, Cre, Cre + N3-L2 SS2-

**e** *mIPSC amplitude*

Cumulative fraction vs Amplitude (pA)

Amplitude (pA): ΔCre, Cre, Cre + N3-L2 SS2-

**f** *IPSCs*

ΔCre (16/3)
Cre (16/4)
Nrxn3α^LNS2-SS2- (15/4)

2 nA / 20 ms

**g** *Evoked IPSC amplitudes*

Peak amplitude (nA) vs Stimulus intensity (µA)

Cre + Nrxn3α^LNS2-SS2-, ΔCre, Cre    ****

**h** *Input/output relationship*

Slope of input/output curves: ΔCre, Cre, Cre + N3-L2 SS2-    **

**i** *IPCS rise times*

Rise time (ms): ΔCre, Cre, Cre + N3-L2 SS2-    *

*IPSC decay times*

Decay time (ms): ΔCre, Cre, Cre + N3-L2 SS2-

**j** *Coeff. of variation*

Coefficient of variation: ΔCre, Cre, Cre + N3-L2 SS2-    **

secondary detection related to low levels of AAV-induced inflammation. Further confirming the efficacy of dystroglycan deletion, qRT-PCR showed a 70% reduction in mRNA levels (S9c). Contrasting prior reports that dystroglycan regulates the number of CCK + inhibitory synapses in the hippocampus[34,42] and maintains inhibitory synapses in

the cerebellum[64], quantifications of the density of inhibitory synapses visualized via immunocytochemistry for gephyrin and synaptoporin failed to detect any change in synapse numbers or size in the external plexiform layer, the area that contains GC→MC synapses (Figs. 8c, d, S9d–i).

**Fig. 5 | Deletion of Nrxn3 in the medial prefrontal cortex (mPFC) impairs inhibitory synaptic function in a manner dependent on alternative splicing at SS2. a** Experimental design for *Nrxn3* deletion following stereotactic infections of the mPFC with AAVs expressing ΔCre (control), Cre, or Cre with the minimal Nrxn3α-LNS2 rescue constructs. **b** Representative fluorescence image of an mPFC section from a mouse infected with AAVs expressing Cre-tdTomato (red) in which a layer 5 pyramidal neuron was patched and filled with neurobiotin (expanded right image; blue). **c**–**e** The *Nrxn3* deletion decreases the mIPSC frequency and amplitude in vivo; however, only the frequency decrease is rescued by the minimal Nrxn3α-LNS2 construct lacking an insert in SS2 (**c**, representative mIPSC traces recorded in the presence of TTX; **d**, **e**: cumulative probability of the mIPSC interevent intervals and amplitudes, insets: summary of the mIPSC frequency and amplitudes). **f**–**h** The *Nrxn3* deletion suppresses the amplitude of IPSCs evoked by extracellular stimulation in layer five and recorded from pyramidal neurons in layer

5 as documented by input/output curves. This phenotype is rescued by the minimal Nrxn3α-LNS2 construct lacking an insert in SS2 (**f**, representative IPSC traces; **g**, summary of input/output amplitudes; **h**, summary of the slope of the input/output curves). **i** The *Nrxn3* deletion increases the rise time of evoked IPSCs (left), but not decay time (right), in a manner that can be rescued by the minimal Nrxn3α-LNS2 rescue constructs. **j** The *Nrxn3* deletion increases the coefficient of variation of IPSCs, suggesting a decrease in release probability, in a manner that can be rescued by the minimal Nrxn3α-LNS2 construct lacking an insert in SS2. Numerical data are means ± SEM; n's (cells/experiments) are indicated above the sample traces and apply to all graphs in an experimental series. Statistical analyzes were performed using two-way ANOVA in **g** and one-way ANOVA in **d**, **e**, **h**–**j** with Dunnett's and Tukey's multiple comparison test with regards to the ΔCre group, with *$p < 0.05$, **$p < 0.01$, and ****$p < 0.0001$. Source data and statistical results for all experiments are provided within the Source Data file.

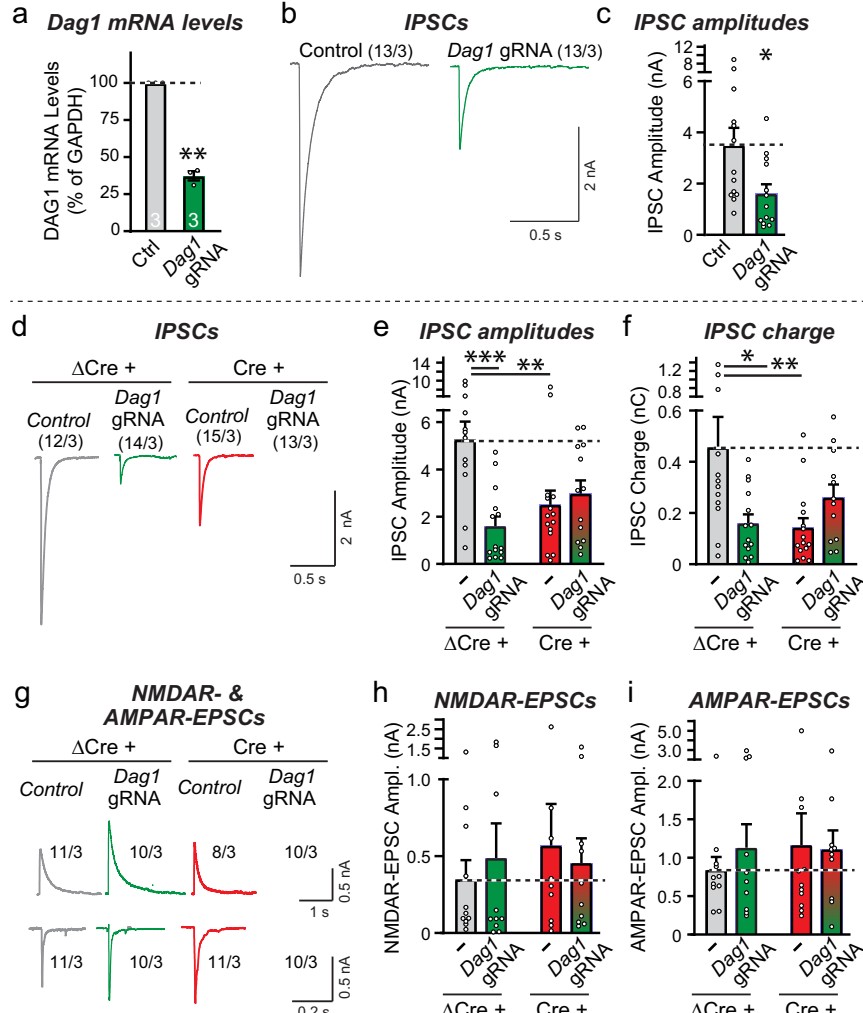

**Fig. 6 | CRISPRi-mediated inhibition of dystroglycan expression in dissociated OB neurons suppresses inhibitory but not excitatory synaptic transmission, with the Nrxn3 and dystroglycan manipulations each occluding the other's phenotype. a**–**c** CRISPR interference (CRISPRi)-mediated inhibition of dystroglycan (*Dag1*) expression decreases the levels of dystroglycan mRNAs and significantly decreases the amplitude of evoked IPSCs (**a**, qRT-PCR measurements of *Dag1* mRNA levels; **b**, sample traces; **c**, summary graph of IPSC amplitudes). **d**–**f** Combined inhibition of dystroglycan (*Dag1*) and *Nrxn3* expression does not lower the evoked IPSC amplitude more severely than the single inhibition of either dystroglycan or of *Nrxn3* expression (**d**, sample traces; **e**, **f**, summary graphs of the IPSC amplitudes (**e**) and charge transfer (**f**)). IPSCs evoked by extracellular stimulation were recorded from mitral/tufted cells in dissociated culture obtained from

*Nrxn3* cKO mice that were infected with lentiviruses expressing either ΔCre and/or the *Dag1* CRISPRi components. **g**–**i** The single or double inhibition of dystroglycan (*Dag1*) and *Nrxn3* expression have no effect on evoked NMDAR- and AMPAR-EPSC amplitudes (**g**, sample traces; **h**, **i** summary graphs of the evoked NMDAR-EPSC amplitudes (**h**) and AMPAR-EPSC amplitudes (**i**)). Experiments were performed as in **d**–**f**. Numerical data are means ± SEM; n's (cells/experiments) are indicated above the sample traces and apply to all graphs in an experimental series. Statistical analyzes were performed with a one-way analysis of variance (ANOVA) with Dunnett's multiple comparison test (**e**, **f**, **h**, and **i**), a two-tailed one sample *t* test (**a**), or a unpaired two-tailed *t* test (**c**), with *$p < 0.05$, **$p < 0.01$, and ***$p < 0.001$. Source data and statistical results for all experiments are provided within the Source Data file.

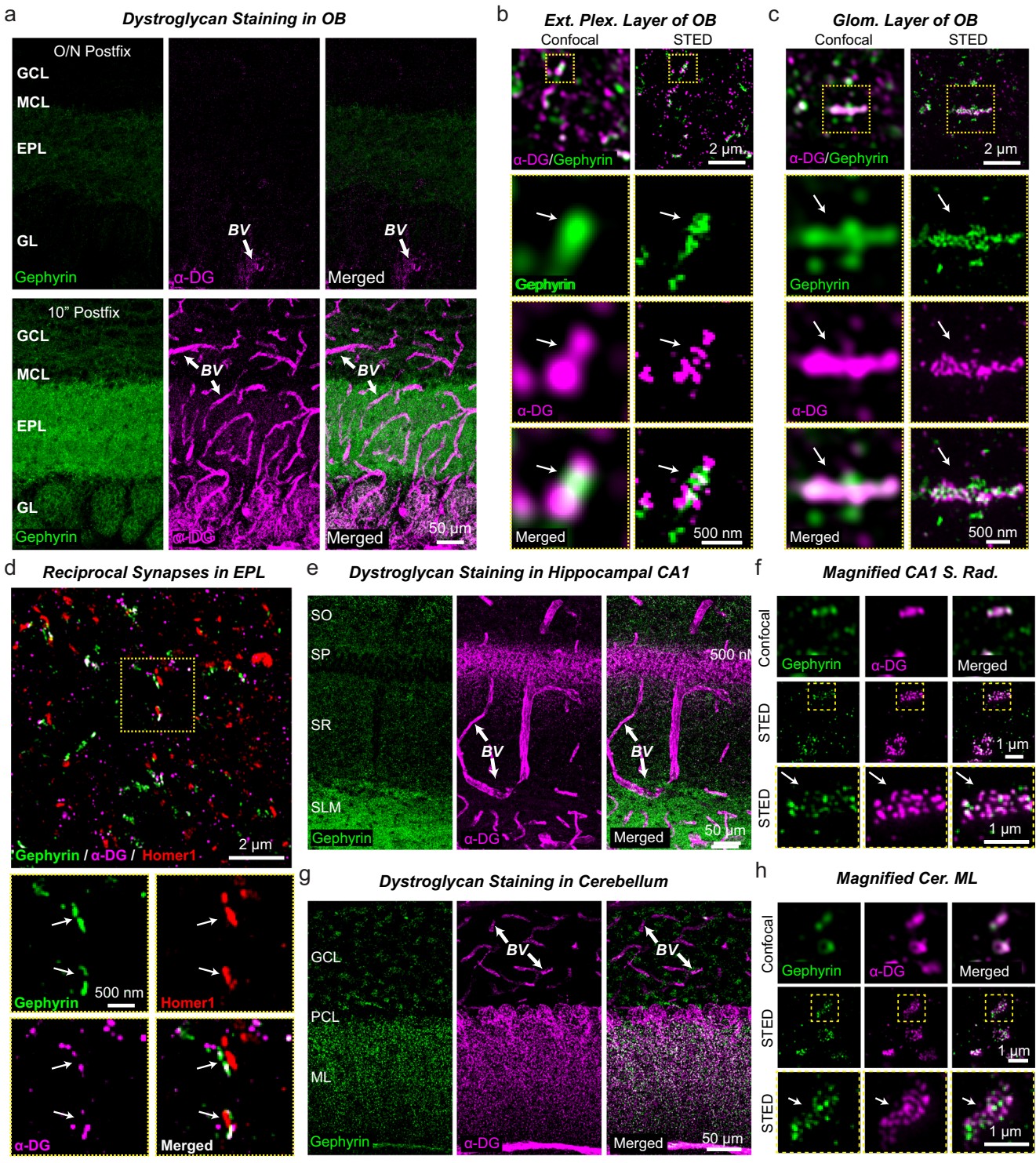

Dystroglycan is expressed not only by neurons but also by astrocytes and pericytes, both of which target dystroglycan to the basal lamina encapsulating blood vessels in the brain (Figs. 7, S8, S9). We next determined whether dystroglycan acts as a postsynaptic ligand in mitral cells to *Nrxn3* expressed by granule cells. To achieve a specifically postsynaptic deletion of dystroglycan in mitral cells, we crossed CAS9 conditional knockin mice[65] with tBet-Cre mice, resulting in the expression of CAS9 only in mitral/tufted cells. Infection of the OB with AAVs expressing the dystroglycan gRNA and Cre-dependent DIO-tdTomato then causes a selective dystroglycan deletion in mitral/tufted cells, with infected mitral cells visualized via their tdTomato expression (Fig. 8e). This enabled us to performed whole-cell patch-

clamp recordings from infected mitral cells in which dystroglycan had been deleted.

We found that the dystroglycan deletion produced a large decrease (~40%) in mIPSC frequency, small but not statistically significant decrease in mIPSC amplitude, and no change in intrinsic electrical properties or mIPSC kinetics (Figs. 8f–h, S9j–l). Importantly, OB slices from male and female mice exhibited similar decreases in mIPSC frequency (Fig. 9m). Moreover, the dystroglycan deletion induced a comparable decrease (~40%) in the amplitude of evoked GC→MC IPSCs (Fig. 8i–k), again without a change in kinetics (Fig. S9n). This decrease in IPSC amplitude was accompanied by a large increase (~80%) in the coefficient of variation of evoked IPSCs (Fig. 8l), and by an

**Fig. 7 | α-Dystroglycan localizes to blood vessels and to inhibitory synapses in the OB. a** Only short post-fixation (10 min) but not overnight fixation (O/N) with 4% PFA allows efficient detection of gephyrin (green) and α-dystroglycan (purple) by immunocytochemistry in the OB (GCL, granule cell layer; MCL, mitral cell layer; EPL, external plexiform layer; GL, glomerular layer). Dystroglycan labeling in blood vessel (BV) walls is indicated with arrows. **b** Super-resolution imaging using stimulated emission depletion microscopy (STED) shows that inhibitory synapses in the EPL of the OB are often co-populated by dystroglycan nanoclusters. **c** STED super-resolution imaging reveals a similar nanocluster structure of gephyrin and dystroglycan at inhibitory synapses in the glomerular layer of the OB. **d** Specific labeling of reciprocal dendrodendritic synapses in OB sections demonstrates the presence of dystroglycan. Dendrodendritic synapses were identified by adjacent localizations of the inhibitory and excitatory postsynaptic markers gephyrin and Homer1, respectively. **e**, **f** Dystroglycan is abundantly present in perisomatic inhibitory synapses of pyramidal neurons in the hippocampal CA1 region in addition to BV walls. Sections were stained for gephyrin and α-dystroglycan (**e**, overview of a CA1 region section [SO, stratum oriens; SP, stratum pyramidale; SR, stratum radiatum; SLM, stratum lacunosome moleculare]; **f**, STED super-resolution imaging of the S. radiatum of the CA1 region demonstrating co-localization of dystroglycan with gephyrin). **g**, **h** Dystroglycan is also present at high levels in inhibitory synapses of the molecular layer of the cerebellar cortex. Sections were stained for gephyrin and α-dystroglycan (**g**, overview of the cerebellar cortex [GCL, granule cell layer; PCL, purkinje cell layer; ML, molecular layer]; **h** STED super-resolution imaging of the molecular layer of the cerebellar cortex again demonstrating co-localization of dystroglycan with the inhibitory synapse marker gephyrin). Experiments were performed at least three times and quantification of dystroglycan association with olfactory inhibitory synapses can be found in Fig. S8.

inversion of paired-pulse suppression to paired-pulse facilitation at short interstimulus intervals (Fig. 8m, n).

The results of the dystroglycan deletion experiments are unexpected in that it was previously argued that dystroglycan is important for the formation and not the operation of a subset of GABAergic synapses, and that its synaptic function does not involve binding to neurexins[34,66]. Therefore we aimed to validate these results in a second set of experiments in which we specifically deleted dystroglycan in mitral cells by infecting the piriform cortex of CAS9 conditional knockin mice with retro-AAVs expressing dystroglycan or control gRNAs and tdTomato (Fig. 9a, b). The retro-AAVs are taken up by axonal projections from the mitral cells to the piriform cortex, resulting in the selective deletion of dystroglycan from mitral cells in the OB without any stereotactic injections of the OB.

Again, the dystroglycan deletion in mitral cells had no apparent effect on inhibitory synapse density or size on mitral cells as examined using immunocytochemistry for the inhibitory synapse marker gephyrin (Fig. 9c, d). The postsynaptic mitral cell deletion of dystroglycan, however, did cause a pronounced functional impairment. Patch-clamp recordings from mitral cells uncovered a robust decrease (~40%) in mIPSC frequency but not amplitude (Fig. 9e–h). No change in intrinsic electrical properties or mIPSC kinetics were present (Fig. S10a–c). Moreover, the dystroglycan deletion greatly decreased (~50%) the amplitude of evoked GC→MC IPSCs (Fig. 9i–k), and increased (~100%) the coefficient of variation of IPSCs without changing the kinetics of the IPSCs (Figs. 9l, S10d). Consistent with this result suggesting a decrease in release probability, the dystroglycan deletion also converted paired-pulse responses from depressed to facilitated at short interstimulus intervals (Fig. 9m, n).

Viewed together, the dystroglycan deletion phenotype is a mirror image of the *Nrxn3* KO phenotype, with a dramatic loss of GC→MC synaptic strength due to a decrease in release probability but without a detectable decrease in synapse numbers. These results are consistent with the observation that rescue of the *Nrxn3* KO phenotype occurs only with Nrxn3α splice variants that bind to dystroglycan. They strongly support the notion that Nrxn3α enables GC→MC synaptic function via binding to dystroglycan. As a final question, we thus asked whether such a mechanism also applies to the role of *Nrxn3* in the mPFC. Indeed, when we applied the CRISPR-mediated deletion of dystroglycan to the mPFC, we also detected a significant decrease in mIPSC frequency without a change in mIPSC amplitude (Fig. 9o–r). Moreover, no major changes in the intrinsic electrical properties or mIPSC kinetics were observed (Fig. S10e–g). Overall, these data support the notion that *Nrxn3* shapes a subset of inhibitory synapses not only in the OB but also in the mPFC by binding to dystroglycan.

## Discussion

Here we show that the binding of presynaptic Nrxn3α to postsynaptic dystroglycan organizes the functional architecture of inhibitory GC→MC synapses in the OB and of inhibitory layer 5/6 synapses in the mPFC (Fig. 10). We demonstrate that the Nrxn3α/dystroglycan interaction is not essential for the formation of these synapses but renders these synapses competent for neurotransmitter release by enabling a normal release probability. Moreover, we find that the role of Nrxn3α at these synapses is controlled by a combinatorial code of alternative splicing whereby SS2 and SS4 of Nrxn3α collaborate to determine the release probability (Fig. 10). Thus, our data propose a molecular feedback mechanism by which binding of presynaptic Nrxn3α to postsynaptic dystroglycan enables Nrxn3α to organize the presynaptic neurotransmitter release machinery. The evidence for these overall conclusions can be summarized as follows:

First, deletion of Nrxn3α lowered the strength of inhibitory neuron→MC synapses in OB cultures by more than a half; this impairment was rescued by Nrxn3α but not by Nrxn3β, with Nrxn3α only being active when its alternatively spliced SS4 and/or SS2 sites contain no insert (Figs. 1, 2; S1–4). SS2 is dominant in this combinatorial splice code because even when SS4 is spliced out, the longer insert in SS2 (SS2ab) blocked the function of Nrxn3α (Fig. 1). Nearly all *Nrxn3* mRNAs in inhibitory OB neurons (~82% of which are granule cells) encode Nrxn3α containing an insert in SS4, and more than 90% of *Nrxn3* mRNAs in inhibitory OB neurons lack in insert in SS2, suggesting that a Nrxn3α/dystroglycan complex is normally favored (Fig. S3, 4). However, it is unknown whether Nrxn3-SS2 and -SS4 alternative splicing may be activity-dependent in these neurons, and this ratio might change during specific behavioral states or during maturation of adult-born OB granule cells, which could regulate GC→MC synaptic transmission by altering the Nrxn3α/dystroglycan interaction.

Second, the mechanism by which deletion of Nrxn3α suppressed GC→MC synaptic transmission consisted of a decrease in the presynaptic release probability, as shown by an increased coefficient of variation of IPSCs, a dramatic shift in paired-pulse ratio, and a lack of change in synapse numbers (Figs. 3–4, S5). Postsynaptic deletion of *Nrxn3* in mitral cells had no effect on GC→MC synaptic transmission (Fig. S6). Thus, this phenotype is similar to the phenotype previously observed following neurexin deficiency in other synapses in which disorganization of calcium channels impairs the coupling of voltage-gated calcium influx to neurotransmitter release[5–7].

Third, the *Nrxn3* KO phenotype at GC→MC synapses is fully rescued by a construct that contains only the LNS2-domain of the extracellular LNS- and EGF-domains of Nrxn3α, provided the LNS2-domain lacks an insert in SS2 (Figs. 2, 4). This rescue was observed both in cultured neurons and in vivo, suggesting that even though the *Nrxn3* deletion causes a decrease in release probability of its resident nerve terminal, a trans-synaptic interaction of Nrxn3α with a postsynaptic trans-ligand is required for GC→MC synapse function. Notably, these findings support a "Swiss Army Knife"-like functional modularity of α-Neurexins, with their large size and presence of an array of independent binding units endowing them with the ability to simultaneously engage diverse trans-synaptic ligands in orchestrating synapse properties.

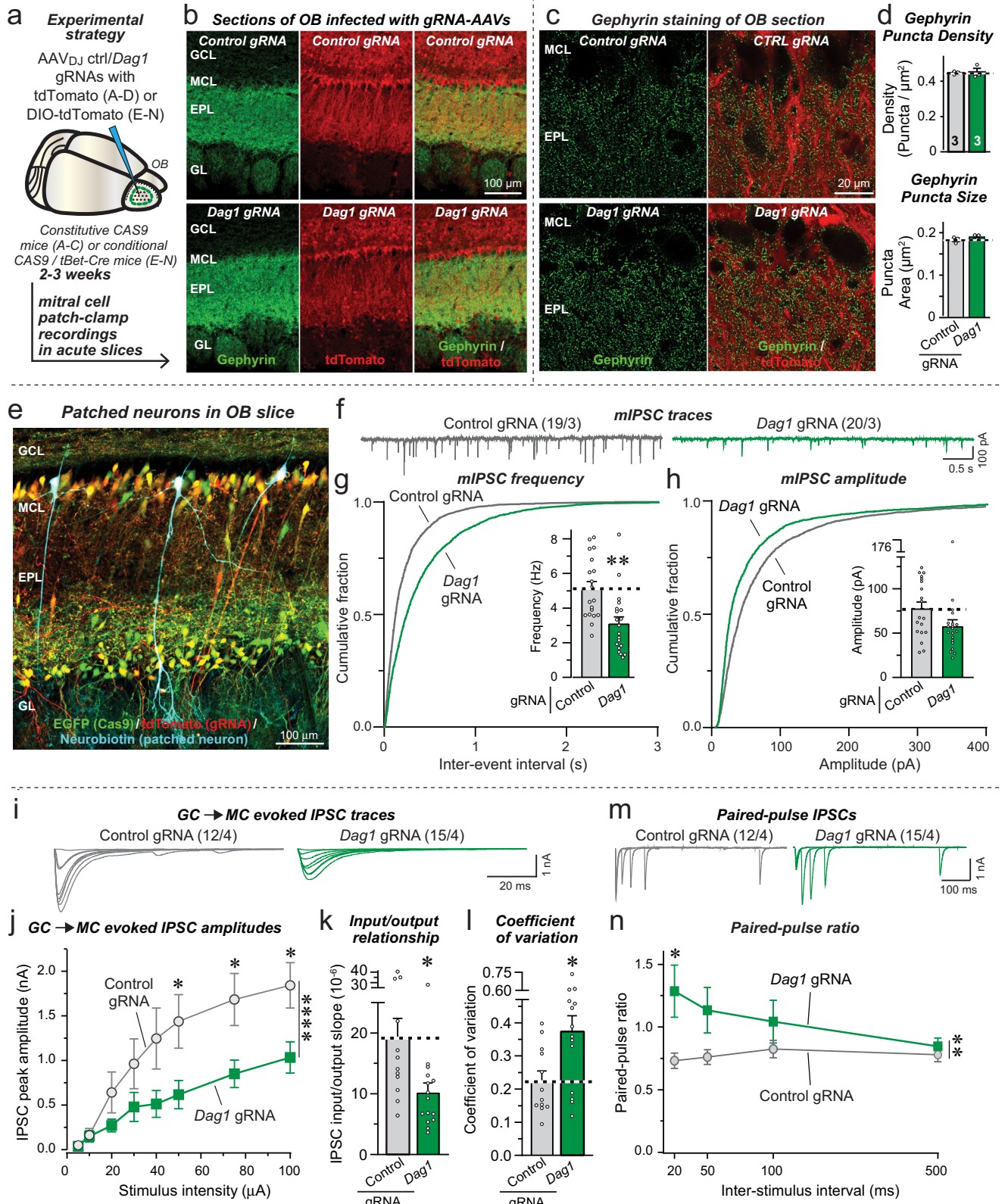

Fourth, the CRISPRi-mediated inhibition of expression and CRISPR-mediated deletion of dystroglycan in postsynaptic mitral cells caused the same phenotype as the presynaptic Nrxn3 deletion in cultured neurons and in vivo (Figs. 6–9). Since the physiological relevance of neurexin-binding to dystroglycan was previously questioned[34], we aimed to confirm this conclusion using two different CRISPR-approaches to delete dystroglycan in vivo from mitral cells, namely direct infection of the OB with AAVs expressing the dystroglycan-specific guide-RNA only in mitral cells (Fig. 8), and retrograde infection of only mitral cells by administration of retro-AAVs expressing the guide-RNA into the piriform cortex (Fig. 9). Importantly, the postsynaptic dystroglycan deletion had no effect on synapse numbers in vivo, but caused the same increase in the coefficient of variation of IPSCs and in their paired-pulse ratio as the presynaptic Nrxn3 deletion.

**Fig. 8 | CRISPR-mediated deletion of dystroglycan (Dag1) in the OB decreases inhibitory GC→MC synaptic transmission by suppressing the release probability. a** Experimental design. The OB of constitutive CAS9-expressing mice was infected stereotactically with AAVs encoding control or *Dag1* gRNAs together with tdTomato at P15-18. Mice were analyzed 2–3 weeks later. **b** Representative fluorescence images of OB sections stained for the inhibitory synapse marker gephyrin (green) and tdTomato expressed via AAVs (red). **c, d** Dystroglycan (*Dag1*) deletion does not change the density or size of gephyrin-positive synaptic puncta (**c**, sample images; **d**, summary of puncta densities (top) and size (bottom)). **e** Representative image of a mitral cell filled with neurobiotin (blue) via the patch pipette (tdTomato expressed via AAVs is shown in red, and EGFP expressed via the CAS9 knockin in green). **f–h** Dystroglycan deletion decreases the mIPSC frequency monitored in mitral cells (**f**, representative mIPSC traces recorded in the presence of TTX; **g**, cumulative probability of the interevent interval and summary of the mIPSC frequency; **h**, cumulative probability and summary of the mIPSC amplitudes).

**i–k** Dystroglycan deletion suppresses inhibitory GC→MC synaptic transmission evoked by extracellular stimulation, as documented by input/output curves (**i**, representative IPSC traces; **j**, summary of input/output amplitudes; **k**, summary of the slope of the input/output curves). **l** Dystroglycan deletion increases the coefficient of variation of evoked IPSCs at GC→MC synapses, suggesting a decrease in release probability. **m, n** *Dag1* deletion induces a large increase in the paired-pulse ratio (**m**, representative traces; **n**, summary of the paired-pulse ratio). Numerical data are means ± SEM; n's (animals (**d**) and cells/experiments (the rest)) are indicated in the summary graph bars (**d**) or above the sample traces (**f**, **i** and **m**) and apply to all graphs in an experimental series with **b–d** belonging to the same series. Statistical analyzes were performed using two-tailed unpaired t-test in **d, g, h, k, l** and two-way ANOVA in **j** & **n** with Bonferroni multiple comparison test, with *$p < 0.05$, **$p < 0.01$, ***$p < 0.001$, and ****$p < 0.0001$. Source data and statistical results for all experiments are provided within the Source Data file.

Given paucity of studies on dystroglycan in the OB compared to other brain regions, we also confirmed widespread localization of dystroglycan at inhibitory synapses in the OB, including reciprocal synapses (Fig. 7). Thus, our findings indicate that postsynaptic dystroglycan binding to presynaptic Nrxn3α retrogradely regulates the presynaptic release probability without affecting synapse formation as such. The strongest evidence for this conclusion comes from the selective rescue of the *Nrxn3* deletion phenotype by Nrxn3α constructs still capable of interacting with dystroglycan as shown previously[26].

Fifth, deletion of *Nrxn3* or of dystroglycan from mPFC neurons produced the same overall phenotype as these deletions induced in OB neurons, namely a loss of inhibitory synaptic strength associated with a change in release probability (Figs. 5, 9). Most importantly, the *Nrxn3* deletion phenotype was again completely rescued by the LNS2-only Nrxn3α construct lacking an insert in SS2 (Fig. 5). The observed phenotype in the mPFC was not as severe as that found in the OB, presumably because we analyzed a relatively homogeneous population of inhibitory GC→MC synapses in the OB in which Nrxn3α/dystroglycan binding invariably shapes synapse properties, whereas we examined a heterogeneous mixture of distinct inhibitory synapses in the mPFC in which only some synapses may utilize the Nrxn3α/dystroglycan signaling mechanism.

Previous work demonstrating that dystroglycan is important for synapses is generally consistent with our results, but most of these studies found a role in synapse formation and/or maintenance instead of synapse function[34,42,64,66]. Since the previous studies were performed in the hippocampus, somatosensory cortex, and cerebellum, while we examined the OB and mPFC, it is possible that the results are due to differences in the type of synapses studied. Moreover, Früh et al. (2016) concluded that the function of dystroglycan in synapses is independent of neurexins, which is plausible since it is a different brain region compared to the region studied here, although neurexins and their binding to the dystroglycan mutant used in the hippocampal studies were not actually examined by Früh et al. (2016). Alternatively, dystroglycan may separately regulate the initial targeting of CCK+ interneurons, which may be the proximal cause of fewer CCK+ synapses in dystroglycan KO mice where dystroglycan was depleted during development[34,42]. At these synapses and others, only once synapses have formed might signaling between dystroglycan and Nrxn3α become critical for sustaining presynaptic release. Another alternative, and possibly most attractive, explanation is that a chronic loss of dystroglycan that impairs inhibitory synapse function may secondarily cause a loss of synapses, which we would have missed in our experiments in which we performed only acute deletions of dystroglycan and *Nrxn3*.

Arguably, our most surprising result is that trans-synaptic binding of presynaptic Nrxn3α to postsynaptic dystroglycan is required for the ability of Nrxn3α to organize a fully functional presynaptic release machinery. What is the nature of the dystroglycan-activated signal in presynaptic terminals – is it a conformational change or dimerization of Nrxn3α or an independent additional signal? This question is likely not only important for understanding the functional molecular architecture of synapses, but also for insight into how mutations in genes associated with dystroglycan, such as mutations in the glycosylating enzymes for dystroglycan or their cytoplasmic binding proteins, and in Nrxn3α produce neurodevelopmental disorders[43–45]. Our findings define the core interaction of Nrxn3α with dystroglycan as functionally essential for inhibitory synapses in at least two brain regions, but they do not yet reveal the detailed molecular signaling that organizes the presynaptic release machinery, a question that will need to be addressed in future.

In summary, we have defined a trans-synaptic signaling complex that performs an indispensable role in enabling a normal presynaptic release probability at a subset of inhibitory synapses. Our findings underscore the notion that individual functional properties of diverse synapses must be systematically studied at a molecular level because information processing in the brain not only depends on how neural circuits are wired via synaptic connections, but also on the functional properties of these connections. Moreover, our current findings add to our understanding of the diverse synaptic roles of neurexins. One might ask why the organization of synapses is so complicated, and why neurexins perform many diverse functions in different types of synapses. This more philosophical question is part of the larger issue of why the brain needs to have many different types of neurons and synapses to operate properly. Naturally, this question is unanswerable at present, but it is striking that with neurexins, a single gene family is used to diversify different types of synapses in the context of distinct circuits. Instead of expressing possibly hundreds of genes to determine synapse identity, with the neurexins the brain expresses only three genes, whose products are uniquely capable of generating thousands of isoforms and of interacting with dozens of ligands. Thereby, the three neurexin genes endow different synapses with distinct properties – a major simplification of the mechanism of synapse diversification. In this view, neurexins do not complicate the design of synapses, but simplify it, even though the overall need for diversity creates a panoply of different molecular pathways whose full extent remains to be characterized.

## Methods
### Animals
Nrxn3 conditional knockout (cKO) mice were generated previously[35] and are available commercially (Jax, 014157). Other mouse lines used in this paper include: tBet-Cre[67], constitutive cas9-knockin (KI) (Jax, 024858), conditional cas9-KI (Jax, 024857), vGAT-Cre (Jax, 028862) and RiboTag mice (Jax, stock# 029977). For analyzing mitral/tufted cell-specific or inhibitory neuron (primarily granule cell) translating mRNA, RiboTag mice were crossed with hemizygous tBet-Cre mice[67] and hemizygous vGAT-Cre mice, respectively. For all experiments

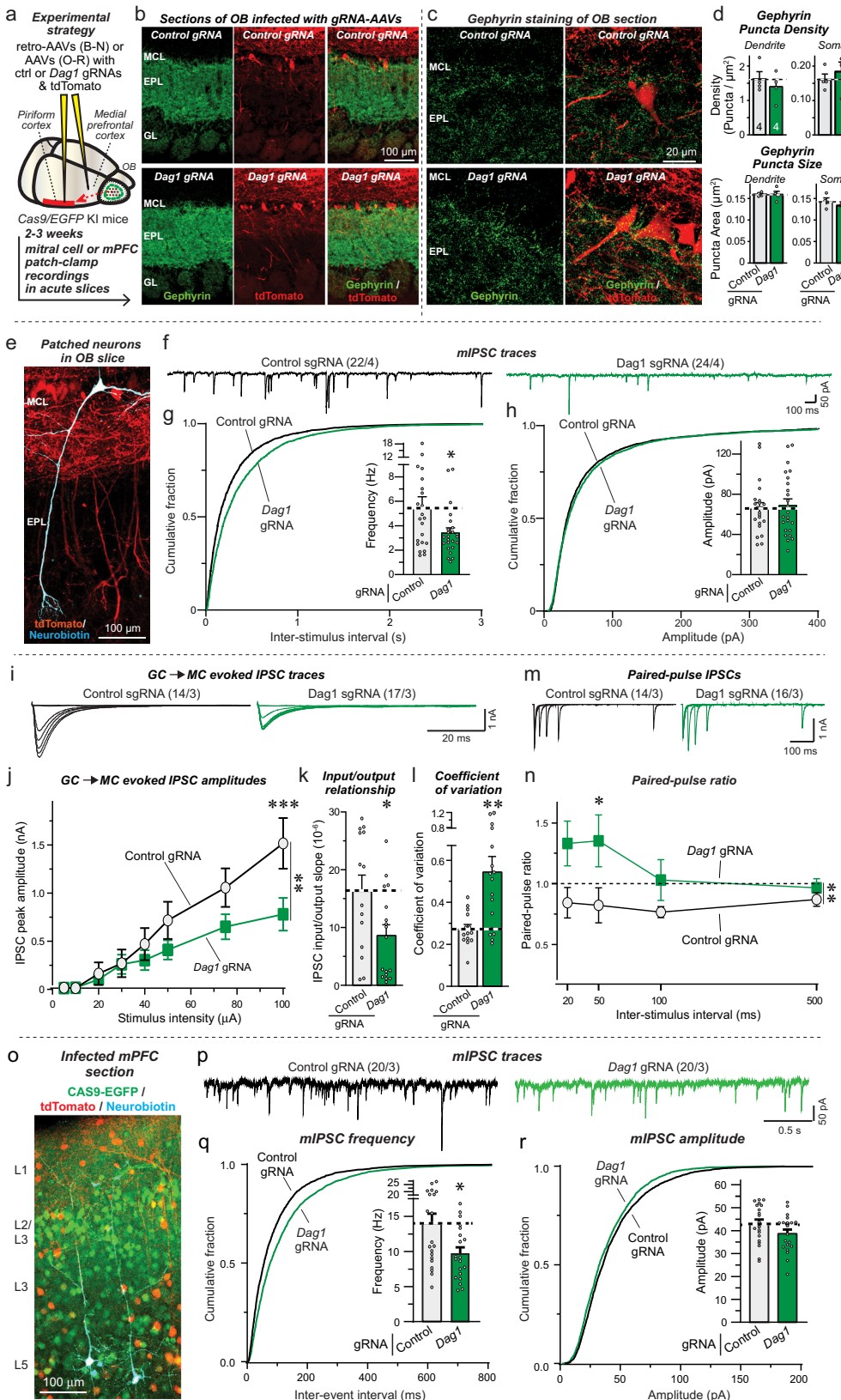

using constitutive and conditional Cas9-KI mice, mice were maintained at homozygosity for the KI alleles. For mitral/tufted cell-specific deletion of *Dag1*, conditional cas9-KI mice were crossed with mice carrying the tBet-Cre allele. Only mice with a single allele of Cre were used for experiments. All mice were weaned at 20 days of age and housed in groups of 2 to 5 on a 12 h light/dark cycle with access to food and water

*ad libidum*. Rooms were maintained with 40–60% humidity and at approximately 22° C. All procedures conformed to National Institutes of Health Guidelines for the Care and Use of Laboratory Mice and were approved by the Stanford Animal Use Committees [Administrative Panel for Laboratory Animal Care (APLAC/) Institutional Animal Care and Use Committee (IACUC)] under the animal protocol 20787.

**Fig. 9 | Dystroglycan (Dag1) deletion in mitral cells of the OB and in the mPFC suppresses inhibitory synaptic transmission. a** Mitral cells are infected with control or *Dag1* gRNAs and tdTomato by projection-specific labeling through retro-AAVs injected in the piriform cortex. **b** Representative OB sections stained for gephyrin (green) and for tdTomato (red). **c, d** Dystroglycan deletion does not change the density or size of gephyrin-positive synapses (green) co-localized with tdTomato-expressing mitral cells (**c**, sample images; **d**, summary of puncta densities (top) and puncta size (bottom)). **e** Representative image of a mitral cell filled with neurobiotin (blue) and expressing tdTomato via AAVs (red). **f–h** Dystroglycan (*Dag1*) deletion decreases the mIPSC frequency in mitral cells (**f**, representative mIPSC traces; **g, h** cumulative probability of the mIPSC interevent intervals and amplitudes, insets: summary of the mIPSC frequency and amplitudes). **i–k** Dystroglycan (*Dag1*) deletion suppresses the IPSC amplitude at GC→MC synapses (**i**, representative IPSC traces; **j**, summary of input/output amplitudes; **k**, summary of the input/output curve slopes). **l** Mitral cell-specific dystroglycan

(*Dag1*) deletion increases the coefficient of variation of evoked IPSCs. **m, n** Dystroglycan (*Dag1*) deletion induces a large increase in the paired-pulse ratio (**m**, representative traces; **n**, summary plot of the paired-pulse ratio). **o–r** Dystroglycan deletion in the mPFC decreases the mIPSC frequency monitored in Layer 5 pyramidal neurons (**o**, representative image of an mPFC section; **p**, representative mIPSC traces; **q–r**, cumulative probability of the mIPSC interevent intervals and amplitudes, insets: summary of the mIPSC frequency and amplitudes). Numerical data are means ± SEM; n's (animals (**d**) or cells/experiments (the rest)) are indicated in the summary graph bars (**d**) or above the sample traces (**f**, **i**, **m**, and **p**) and apply to all graphs in an experimental series with **b–d** belonging to the same series. Statistical analyzes were performed using two-tailed unpaired *t*-test in **d**, **g**, **h**, **k**, **l**, **q**, **r**, and by two-way ANOVA in **j** & **n** with Bonferroni multiple hypothesis testing, with *$p < 0.05$, **$p < 0.01$, and ***$p < 0.001$. Source data and statistical results for all experiments are provided within the Source Data file.

## Plasmids

Lentiviral vectors for expression of Cre and ΔCre (truncated, inactive) recombinase driven by the human synapsin-1 promoter have been described previously[68]. For all other experiments using the lentiviral backbone with a human synapsin-1 vector (i.e. FSW), an empty vector was used as a control. For all Nrxn3 rescue constructs, a single HA tag was positioned between the native signal peptide and was flanked by linker sequences (i.e. glycine-glycine-serine upstream and glycine-serine downstream). All culture rescue constructs were incorporated into the FSW lentiviral backbone. A library of previously published cDNA's[12,35] were used to clone Nrxn3alpha and Nrxn3beta splice variants described in Figs. 1 and 2. For all truncation constructs (Fig. 2), that lacked LNS6, upstream domains were fused at the same position that LNS6 would normally be, thus preserving the downstream stalk region, transmembrane domain, and cytoplasmic sequence.

The adeno-associated virus (AAV) serotypes used in this study were AAV-DJ and rAAV2-retro for retrograde experiments[69]. AAV backbones were generated to allow the expression of Cre and ΔCre fused to tdTomato, minimal Nrxn3 LNS2 rescues (with and without SS2), and gRNA's targeting *Dag1* with soluble tdTomato driven by the hSynI promoter in a Cre-sensitive (i.e. with DIO) or constitutive manner. For CRISPRi lentiviral backbones, a scrambled gRNA control was generated (5'-GCGCCAAACGTGCCCTGACG-3'). For targeting dystroglycan, several gRNA's were initially screened. The final gRNA that performed best following a functional screen was 5'-AGCTTCGCGCGGAGTCCCCG-3'. CRISPRi was performed using a lentiviral backbone described previously[70], with an expression of gRNA driven by the U6 promoter and the inactive Cas9 fused to KRAB driven by the EFS promoter.

For in vivo CRISPR experiments, two gRNA were used to ensure efficient targeting of *Dag1* including one driven by a U6 promoter (i.e. 5'-tggttaggttctccccccacg-3') and another by a H1 promoter (i.e. 5'-accgtggttggcattccaga-3'). These gRNA were published previously[41]. Scrambled gRNA sequences were used as controls.

Sequences for all unpublished constructs are included in a supplementary information file.

## Antibodies

The following antibodies were used at the indicated concentrations (IHC-immunohistochemistry; ICC-immunocytochemistry): anti-alpha-Dystroglycan [45–3] rabbit (Abcam Cat# ab199768; 1:250 IHC), anti-Dystroglycan (Millipore Cat#05-593; 1:250 IHC), anti-HA rabbit (Cell Signaling Cat# 3724; 1:500 IHC), anti-Gephyrin mouse (Synaptic Systems Cat# 147 011; 1:1000 ICC), anti-Gephyrin guinea pig (Synaptic Systems Cat# 147 318; 1:250 IHC), anti-Homer1 rabbit (Synaptic Systems Cat# 160 003, 1:1000 ICC/IHC), anti-MAP2 chicken (Encorbio Cat# CPCA-MAP2; 1:1000 ICC), anti-GABA$_A$Rα1 (Synaptic Systems Cat# 224 203; 1:250 live ICC), anti-GABA$_A$Rα2 (Synaptic Systems Cat# 224 103; 1:250 live ICC), anti-GABA$_A$Rγ2 (Synaptic Systems Cat# 224 003;

1:250 live ICC), anti-Synaptophysin-2 rabbit (homemade, Wang et al., 2021; 1:500 IHC), and anti-vGAT guinea pig (Synaptic Systems Cat# 131 004; 1:1000 ICC), Goat anti-Mouse IgM Heavy Chain Alexa594 (ThermoFisher, A-21044; 1:400, IHC/STED), Goat anti-Mouse IgG Alexa546 (ThermoFisher, A11003; 1:1000, ICC/IHC), Goat anti-Mouse IgG Alexa405 (ThermoFisher, A31553; 1:1000, ICC/IHC), Goat anti-Rabbit IgG Alexa405 (ThermoFisher, A31556; 1:1000, ICC/IHC), Goat anti-Rabbit IgG Alexa546 (ThermoFisher, A11010; 1:1000, ICC/IHC), Goat anti-Rabbit IgG STAR Red (Abberior, STRED-1001; 1:400, IHC), Goat anti-Rabbit IgG STAR460L (Abberior, ST460L-1002; 1:400, IHC), Goat anti-Rabbit IgG CF568 (Biotium, 20098-1 mg; 1:3000, IHC), Goat anti-Guinea Pig IgG Alexa647 (ThermoFisher, A21450; 1:1000-1:3000, ICC/IHC), Goat anti-Guinea Pig IgG STAR Red (Abberior, STRED-1006; 1:400, IHC), and Goat anti-Chicken Igy Alexa488 (ThermoFisher, A11039; 1:1000, ICC).

## Cell culture

**Primary neuron cultures (containing glia).** Hippocampal, OB, and cortical neurons were cultured from newborn mice. Tissue was dissected and mixed regardless of sex. In general, pooling tissue from three to six mice in a given preparation was used to generate cultures. Cortical neurons were derived from entire cortical lobes that were separated from midbrain, hindbrain, and hippocampus. Olfactory bulbs were plate at 4 coverslips, in a 24-well dish, per mouse (2 bulbs). Dissected hippocampi, OBs, or cortices were digested for 20 min with 10 U/ml papain in Hank's buffered saline (HBS) in an incubator, washed with HBS, dissociated in plating media (MEM supplemented with 0.5% glucose, 0.02% NaHCO$_3$, 0.1 mg/ml transferrin, 10% FBS, 2 mM L-glutamine, and 0.025 mg/ml insulin), and seeded on Matrigel (BD Biosciences) precoated coverslips placed inside 24-well dishes. For OB neurons, the day after plating, 95% of media was replaced with MEM (GIBCO) supplemented with 2% B27 (GIBCO), 0.5% w/v glucose, 100 mg/l transferrin, 5% fetal bovine serum. For hippocampal and cortical neurons, the day after plating, 95% of the plating medium was replaced with neuronal growth medium lacking serum (Neurobasal-A medium supplemented with 2% B27 supplement and 0.5 mM L-glutamine). At DIV2–3 (for hippocampal and OB cultures) or DIV3–4 (for cortical cultures), 50% of the medium was exchanged with fresh growth medium additionally supplemented with 4 μM AraC (Sigma-Aldrich) to restrict glial overgrowth. When applicable, neurons were infected between DIV3-4 with lentiviruses expressing EGFP-tagged ΔCre (control) or Cre without and/or with the indicated rescue constructs driven by the synapsin promoter. For long-term culture of hippocampal and cortical neurons, 25% fresh media was added every 4–5 d starting from DIV7. A partial media change (<30%) was performed only once on DIV7 for OB neurons to preserve cell health. OB cultures do not maintain cultural health if media is exchanged too frequently and if the exchange volume exceeds 30%.

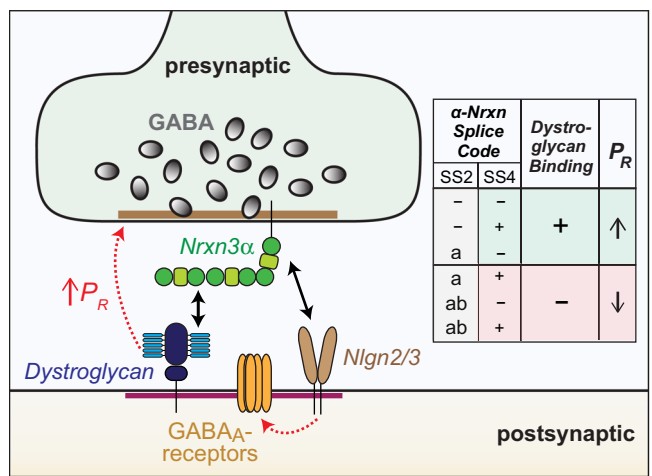

| | α-Nrxn Splice Code | | Dystro-glycan Binding | $P_R$ |
|---|---|---|---|---|
| SS2 | SS4 | | | |
| − | − | | | |
| − | + | | + | ↑ |
| a | − | | | |
| a | + | | | |
| ab | − | | − | ↓ |
| ab | + | | | |

**Fig. 10 | Model of transsynaptic synaptic signaling by Nrxn3α and dystroglycan at inhibitory synapses.** Summary cartoon of a complex between presynaptic Nrxn3α and postsynaptic dystroglycan and Neuroligin-2 and -3 (Nlgn2/3). Right, summary of the α-neurexin splice code that dictates dystroglycan binding, which in turn retrogradely elevates the presynaptic release probability ($P_R$).

**HEK293T cells.** HEK293T cells were purchased from American Type Culture Collection (CRL11268). The cells were isolated from human embryo kidney tissue and this particular line is a derivative of the 293T (293tsA1609neo) cell line. Cells were grown in complete DMEM (cDMEM), which consisted of DMEM (Gibco), 5% FBS (Sigma), penicillin, and streptomycin. All transfections were performed using lipofectamine 3000 (Invitrogen). For co-culture assays, HEK293T cells were plated on 12-well plates and transfected at ~90% confluency according to the manufacturer's instructions.

### Preparation of viral particles

**AAV preparation.** The adeno-associated virus (AAV) serotypes used in this study were AAV-DJ and rAAV2-retro for retrograde experiments[69]. HEK293T cells were transfected with the helper plasmid, the serotype-specific plasmid, and the AAV plasmid using homemade calcium phosphate solution. Cells were dissociated and precipitated 72 hours post-transfection. Nuclei were lysed by three times of freeze-thaw cycles and were later treated with Benzonase nuclease (Sigma-Aldrich, cat # E1014). The supernatant then underwent iodixanol gradient ultracentrifugation for 3 hours at 273720.9 x g at 4°C in a S80AT3 rotor. AAV were then concentrated using filtered centrifugation and dialyzed in minimal essential media (MEM).

**Lentivirus preparation.** Recombinant lentiviral particles were produced in HEK293T cells by co-transfecting cells with long terminal repeat (LTR) containing vector and helper plasmids (pRSV-REV, pMDLg/gRRE, and pVSVG) using calcium phosphate. Media were exchanged 1 h before transfection and included 25 μM chloroquine diphosphate. Per 75 cm2 of cells, 0.5 ml of 250 mM CaCl2 containing molar equivalents of DNA (12 μg of LTR-containing vector, 3.9 μg pRSV-REV, 8.1 μg pMDLg/gRRE, and 6.0 μg pVSVG) was added dropwise to an equal volume of 2X-HBS (0.4 M NaCl, 10 mM KCl, 1.5 mM Na2HPO4, 0.2% glucose, and 38.4 mM HEPES, pH 7.05) under vigorous mixing, incubated for 20 min at room temperature, and added dropwise to the cells. A total of 16–20 h following transfection, cells were washed with plain DMEM and replaced with neuronal growth media lacking AraC. After 24 h, media containing lentiviral particles were cleared by centrifugation (1500 x *g*, 10 min), aliquoted, and snap-frozen. Neuronal cultures were infected with lentivirus on DIV3 or 4 by adding 25–30 μl of viral supernatant per well of a 24-well plate.

### Electrophysiology

**Culture electrophysiology.** Cultured neurons were collected and recorded at DIV 14–17. Electrophysiology recordings were performed at room temperature, performed in whole-cell patch-clamp mode using concentric extracellular stimulation electrodes as described previously[71]. The glass pipettes (2−3 MΩ filled with intracellular pipette solution) were pulled from borosilicate glass capillaries with a vertical micropipette puller (PC-10, Narishige). After the formation of the whole-cell configuration and equilibration of the intracellular pipette solution, the series resistance was adjusted to 8−10 MΩ. Synaptic currents were monitored with a Multiclamp 700B amplifier (Molecular Devices). A bipolar stimulation electrode (FHC, Bowdoinham, ME) was placed 100-150 μm from the soma of the neurons recorded to apply focal square pulse stimuli (duration 1 ms) and trigger evoked synaptic responses. The frequency, duration, and magnitude of the extracellular stimulus were controlled with a Model 2100 Isolated Pulse Stimulator (A-M Systems) synchronized with Clampex 9 data acquisition software (Molecular Devices). The whole-cell pipette solution contained (in mM): 120 CsCl, 5 NaCl, 1 MgCl2, 10 HEPES, 10 EGTA, 0.3 Na-GTP, 3 Mg-ATP, and 5 QX-314 (pH 7.2, adjusted with CsOH). The bath solution contained (in mM): 140 NaCl, 5 KCl, 2 MgCl2, 2 CaCl2, 10 HEPES, and 10 glucose (pH 7.4, adjusted with NaOH). IPSCs, as well as AMPAR- or NMDAR-mediated EPSCs, were pharmacologically isolated by adding blockers against the AMPA receptor (CNQX, 10 μM), NMDA receptor (APV, 50 μM), or GABA$_A$ receptor (picrotoxin, 50 μM) to the extracellular solution. Spontaneous mIPSCs and mEPSCs were monitored in the presence of tetrodotoxin (TTX, 1 μM) to block action potentials. Miniature events were analyzed in Clampfit 9 and 10.7 (Molecular Devices) using the template matching search and a minimal threshold of 5 pA and each event was visually inspected for inclusion or rejection by an experimenter blind to the recording condition.

**Slice electrophysiology.** Two to three weeks after viral injection, mice were anesthetized via isoflurane inhalation, and brains were quickly dissected. The dissected brain was sliced in ice-cold oxygenated (95% O₂ and 5% CO₂) cutting solution (228 mM sucrose, 11 mM glucose, 26 mM NaHCO₃, 1 mM NaH₂PO₄, 2.5 mM KCl, 7 mM MgSO₄, and 0.5 mM CaCl₂). Horizontal sections for OB and coronal sections for mPFC, both of which were 300 μm thick, were obtained by using a vibratome. Slices were quickly transferred to oxygenated artificial cerebrospinal fluid (ACSF; 119 mM NaCl, 2.5 mM KCl, 1 mM NaH₂PO₄, 1.3 mM MgSO₄, 26 mM NaHCO₃, 10 mM glucose, and 2.5 mM CaCl₂) at 32 °C for 30 min. Slices were allowed to recover at room temperature for an additional 30 min. The recording chamber was temperature controlled and set to 32 °C, and ACSF was perfused at 1 mL/min. The internal solution for whole-cell patch clamp contained 135 mM CsCl, 10 mM HEPES, 1 mM EGTA, 1 mM Na-GTP and 4 mM Mg-ATP pHed to 7.4. 10 mM QX314-bromide was added for evoked recordings. 0.2% neurobiotin (VectorLab) was included for morphological reconstruction. The pipette resistance ranged from 1.8 to 2.5 MΩ. Mitral cells were identified in the mitral cell layer in the OB and mPFC neurons were identified either by the fluorescent reporter or pyramidal-shaped neuron in the deep layer. Access resistance was under 10 MΩ (for mitral cells) and 15 MΩ for mPFC neurons throughout the experiment. 1 μM TTX (Tocris), 20 μM CNQX (Tocris), and 50 μM D-AP5 (Tocris) were included in the bath for mIPSC. Twenty micrometers of CNQX (Tocris) and 50 μM D-AP5 (Tocris) were included in the bath for evoked IPSC recordings. All recordings were done in voltage-clamp mode with a holding potential of −70mV. For eIPSC stimulation, concentric bipolar electrode was used. For GC→MC eIPSC, the stimulating electrode was placed directly below the mitral cell with constant distance roughly at the junction between the internal plexiform layer and granule cell layer, 30 μm below the surface of the slice. For eIPSC in mPFC, the stimulating electrode was placed parallel to the recorded cell in the same layer with constant distance and 30 μm below the

surface of the slice. The experimenter was blind to the treatment groups during recordings and analysis.

## Stereotactic injections

Mice were prepared for stereotactic injections using standard procedures approved by the Stanford University Administrative Panel on Laboratory Animal Care. Mice were anesthetized by 0.2 mL avertin working solution per 10 grams body weight. The avertin stock solution was made by dissolving 5 grams tribromoethanol into 5 mL T-amyl alcohol, which was further diluted 80 folds in DPBS to make the avertin working solution. The coordinates (AP/ML/DV from Bregma) and volumes for the intercranial injections are (1) +4.3/±0.85/−1.7 and +5.3/±0.6/−1.5 with 1.0 uL virus for the OBs and (2) −0.7/±3.7/−4.75 with 0.75 uL virus for the piriform cortex (for retrograde targeting of mitral cells). For injection into the piriform cortex, the mouse brains were aligned to have less than 0.05 mm difference on the DV axis at −1.00 (AP) between ±3.00 (ML) positions. The reference zero point for DV is on the surface of the OB for OB injection and the surface of the skull at AP/ML of −0.7/0.0 for piriform cortex injection.

## Purification of tissue mRNA

Wild-type (CD1) mice at 8 weeks of age were euthanized using isoflurane and decapitated. Several brain regions including the cortex, OB, cerebellum, hippocampus, and pons/medulla were quickly dissected and snap-frozen in liquid nitrogen or dry ice and transferred to −80 °C storage until processing. The specimen was subjected to RNA extraction using the QIAGEN RNeasy Micro kit.

## Purification of ribosome-bound mRNA

RiboTag mice[72] were crossed with tBet-Cre mice[67] or vGAT-Cre mice. After OB extraction described above, the frozen bulbs were partially thawed in fresh homogenization buffer at 10% (w/v) and Dounce homogenized. Homogenates underwent centrifugation, and 10% of the supernatant was used as input. The remaining supernatant was incubated with prewashed anti-HA magnetic beads (Thermo Fisher Scientific) overnight at 4 °C. The beads were washed three times with a high-salt buffer followed by elution with RLT lysis buffer containing 2-mercaptoethanol. The sample and the input were then subjected to mRNA extraction described above. RNA concentration was determined using a NanoDrop 1000 Spectrophotometer (Thermo) and stored at -80˚C until downstream analysis.

## qPCR

Quantitative reverse transcription (RT)–PCR was performed in triplicates for each condition with QuantStudio 3 (Thermo Fisher Scientific). RNA (20 ng) was used for each reaction, in conjunction with TaqMan Fast Virus 1-Step Master Mix (Thermo Fisher Scientific) and gene-specific qRT-PCR probes [IDT (integrated DNA technologies). Predesigned PrimeTime qPCR probe assays (IDT) were used for vGluT1 (Mm.PT.58.12116555), vGaT (Mm.PT.58.6658400), aquaporin-4 (Mm.PT.58.9080805), MBP (Mm.PT.58.28532164), ActB (Mm.PT.39a.22214843.g), Cbln1 (Mm.PT.58.12172339), Cbln2 (Mm.PT.58.5608729), Cbln4 (Mm.PT.58.17207498), Grid1 (Mm.PT.58.32947175), and Grid2 (Mm.PT.58.12083939), Nlgn1 (Mm.PT.58.30240881), Nlgn2 (Mm.PT.58.16799702), Nlgn3 (Mm.PT.58.31138258, Nxph1 (Mm.PT.58.13767897), Nxph2 (Mm.PT.58.28481365), Nxph3 (Mm.PT.12688150), Nxph4 (Mm.PT.58.11246838), LRRTM1 (Mm.PT.58.42587284.g) LRRTM2 (Mm.PT.58.6337058.g), LRRTM3 (Mm.PT.58.31131475), LRRTM4 (Mm.PT.58.11146838), Fam19a1 (Mm.PT.56a.6079538), Fam19a2 (Mm.PT.58.7298614), Fam19a4 (Mm.PT.56a.9330679), Car10 (Mm.PT.58.11765793), Car11 (Mm.PT.58.32895602), Dag1-ex1/ex2 (Mm.PT.58.46076316), Dag1-ex3/ex4 (Mm.PT.58.45967735), and Dag1-ex4/ex5 (Mm.PT.58.5524327). Customed PrimTime qPCR probe assays (IDT) were used for Nrxn1α (forward: TTCAAGTCCACAGATGCCAG;

reverse: CAACACAAATCACTGCGGG; probe: TGCCAAAACTGGTCCATGCCAAAG), Nrxn1β (forward: CCTGTCTGCTCGTGTACTG; reverse: TTGCAATCTACAGGTCACCAG; probe: AGATATATGTTGTCCCAGCGTGTCCG), Nrxn1γ (forward: GCCAGACAGACATGGATATGAG; reverse: GTCAATGTCCTCATCGTCACT; probe: ACAGATGACATCCTTGTGGCCTCG), Nrxn2α (forward: GTCAGCAACAACTTCATGGG; reverse: AGCCACATCCTCACAACG; probe: CTTCATCTTCGGGTCCCCTTCCT), Nrxn2β (forward: CCACCACTTCCACAGCAAG; reverse: CTGGTGTGTGCTGAAGCCTA; probe: GGACCACATACAT CTTCGGG), Nrxn3α (forward: GGGAGAACCTGCGAAAGAG; reverse: ATGAAGCGGAAGGACACATC; probe: CTGCCGTCATAGCTCAGGATAGATGC), Nrxn3β (forward: CACCACTCTGTGCCTATTTC; reverse: GGCCAGGTATAGAGGATGA; probe: TCTATCGCTCCCCTGTTTCC), Nlgn1 (forward: GGTTGGGTTTGGTATGGATGA; reverse: GATGTTGAGTGCAGTAGTAATGAC; probe: TGAGGAACTGGTTGATTTGGGTCACC), Nlgn2 (forward: CCGTGTAGAAACAGCATGACC; reverse: TGCCTGTACCTCAACCTCTA; probe: TCAATCCGCCAGACACAGATATCCG), and Nlgn3 (forward: CACTGTCTCGGATGTCTTCA; reverse: CCTCTATCTGAATGTGTATGTGC; probe: CCTGTTTCTTAGCGCCGGATCCAT).

Assays generating $C_t$ values >35 were omitted. $C_t$ values for technical replicates (duplicate or triplicate) differed by less than 0.5. $C_t$ values were averaged for technical replicates. Data were normalized to the arithmetic mean of *ActB* and *Gapdh* using the $2^{-\Delta\Delta Ct}$ method.

## Junction-flanking PCR

The following primers anneal to constitutive exon sequences that flank splice junctions and thus amplify *Nrxn1-3* mRNA transcripts with or without alternative splice sequences (splice site, forward primer, reverse primer):

Nrxn1-SS2v1 (5′-TGGGATCAGGGGCCTTTGAAGCA-3′, 5′-GAAGGTCGGCTGTGCTGGGG-3′), Nrxn2-SS2v1 (5′-GCACGACGTCCGGGTTACCC-3′, 5′-GGTCGGCTGTGTTGGGGCTG-3′), Nrxn3-SS2v1 (5′-TCCGGGGCCTTTGAGGCCAT-3′, 5′-GCGGTACTTGGGCTTCCACCA-3′), Nrxn1-SS4v1 (5′-CTGGCCAGTTATCGAACGCT-3′, 5′-GCGATGTTGGCATCGTTCTC-3′), Nrxn2-SS4v1 (5′-CAACGAGAGGTACCCGGC-3′, 5′-TACTAGCCGTAGGTGGCCTT-3′), Nrxn3-SS4v1 (5′-ACACTTCAGGTGGACAACTG-3′, 5′-AGTTGACCTTGGAAGAGACG-3′), Nrxn1-SS2v2 (5′-TGCCTGGCATGATGTGAA-3′, 5′-TGGTGTAATCTTCTTGCGTGTA-3′), Nrxn2-SS2v2 (5′-ACCCGTCAATGGCAAGTT-3′, 5′-AGCCCAGCATGGTGTAATC-3′), Nrxn3-SS2v2 (5′-CCTGGCATGATGTCAAAGTG-3′, 5′-GCCCAGCATGGTGTAGT-3′), Nrxn1-SS4v2 (5′-CCAGTTATCGAACGCTACCC-3′, 5′-GCCATTGTAGTAAAGACCAGAGA-3′), Nrxn2-SS4v2 (5′-GACAGCTGGCCAGTCAAC-3′, 5′-GACACCTGGCCCTGGAA-3′), Nrxn3-SS4v2 (5′-GGCCAGTGAATGAGCACTAT-3′, 5′GACACCTGGCCCTGGAA-3′). Fig S3a–b use Nrxn1/2/3-SS4v1 and Nrxn1/2/3-SS2v1. Fig. S3c–d use Nrxn1/2/3-SS4v2 and Fig. S4c–d use Nrxn1-SS2v1 and Nrxn2/3-SS2v2. Fig. S3h use Nrxn1/2/3-SS4v1 and Figs. S4e–f use Nrxn1/2/3-SS2v1.

cDNA was synthesized from equal amounts of 1) adult brain regions, 2) primary neuron/glia culture mRNA, or 3) immunoprecipitated mRNA from mitral/tufted cells or granule cells and total input mRNA from the OB. Junction-flanking PCR was then performed with equal amount of cDNA from groups being compared. The PCR products were separated on homemade MetaPhor agarose gel (Lonza) and stained with GelRed. Stained gel was imaged at sub-saturation using the ChemiDoc Gel Imaging System (Bio-Rad). Quantification was performed using Image Lab (Bio-Rad) or ImageStudioLite (LI-COR). Intensity values were normalized to the size of DNA products to control for intensity differences caused by different dye incorporation owing to varied DNA length. For an example of presentation of full scan blots in supplementary Figs. 3–4, see the Supplementary Information file.

## Immunocytochemistry

For live surface-labeling experiments, primary neurons were first washed at room temperature once with a HEPES bath solution, which

contained the following (in mM): 140–150 NaCl, 4–5 KCl, 2 $CaCl_2$, 1 $MgCl_2$, 10 glucose, and 10 HEPES, with pH adjusted to 7.4 with NaOH, and osmolarity of 300 mOsm. Cultures were then incubated at room temperature for 20 min with antibodies recognizing the $GABA_AR\alpha1$, $GABA_AR\alpha2$, or $GABA_AR\gamma2$ diluted in HEPES bath solution. Cultures were then gently washed three times with HEPES bath solution, followed by fixation for 20 min at room temperature with 4% (wt/vol) PFA. Following fixation, cultures were washed three times with Dulbecco's PBS (DPBS). Cultures were blocked for 1 h at room temperature with antibody dilution buffer (ADB) without Triton X-100(−), which contains 5% normal goat serum diluted in DPBS. Cells were then labeled with Alexa Fluor–conjugated secondary antibodies (1:1,000; Invitrogen) diluted in ADB(−) for 2 h at room temperature. Cultures were then incubated for 10 minutes with 4% PFA for post-fixation and stains proceeded as described above.

For all immunocytochemistry experiments, cells were washed and then permeabilized and blocked for 1 h with ADB with tx-100(+) which contains 0.3% tx-100 and 5% normal goat serum diluted in DPBS. Non-surface primary antibodies were diluted in ADB(+) and cells were incubated in the cold-room overnight or for 2 h at RT. Cultures were washed three times with DPBS and then incubated with Alexa Fluor-conjugated secondary antibodies (1:1000; Invitrogen) diluted in ADB(+) for 1 h at RT. After three additional washes, coverslips were inverted onto glass microscope slides with Fluoromount-G mounting media (Southern Biotech).

### Immunohistochemistry

Mice were anesthetized with isoflurane and then transcardially perfused (~1 ml/min) for 1 min with 0.1 M DPBS (RT) followed by 7 min with 4% PFA (Electron Microscopy Services). For synaptic protein quantification using confocal microscopy, OBs were dissected and post-fixed for 13 minutes at RT. For imaging of the hippocampus and cortex, brains were post-fixed for 2 h. For STED super resolution microscopy, OB's were post-fixed for 10 min at RT, 20 min at RT, and O/N at 4 °C. Following fixation, tissue was washed 3 times with DPBS and cryoprotected by a 24–48 h incubation in 30% sucrose w/v in DPBS. Tissue was embedded in OCT Compound (Sakura), sectioned on the sagittal plane at 30 μm using a cryostat, and stored as floating sections in DPBS. For staining, free-floating sections were incubated with blocking buffer (containing 5% NGS, 1% Pen-Strep, and 0.5% tx-100 in DPBS) for 1 h at RT. Sections were then incubated with primary antibodies diluted in blocking buffer overnight at RT on a rocker with slight agitation. After 3 washes, sections were incubated with Alexa dye secondary antibodies diluted in blocking buffer for 1–2 h at RT. For STED imaging, secondary antibodies raised in goat were conjugated to Abberior STAR RED, STAR ORANGE, and 460 L and used at 1:400 dilution. For 2-color STED, only STAR RED and STAR ORANGE secondary antibodies were used. Sections were washed 3–4 times (with 0.05% tx-100 in PBS) and then mounted on charged glass slides. After drying, sections were dipped in water and allowed to dry again. For confocal imaging, per slide, 4 droplets of Fluormount-G with or without DAPI was added, slides were coverslipped, and nail polish was used to secure the coverslip until mounting medium hardened. For STED imaging, sections were mounted on poly-l-lysine coated coverslips prior to being mounted on a slide using abberior MOUNT, SOLID according to manufacturer's instructions. Slides were allowed to solidify for at least 24 h prior to imaging.

### Confocal microscopy

All confocal images were acquired at RT using an inverted Nikon A1RSi confocal microscope equipped with a 20x, 60x, or 100x objective (Apo, NA 1.4) and operated by NIS-Elements AR acquisition software. For quantitative analysis of synaptic puncta, high magnification images were taken at 1024 × 1024 pixels with a z-stack distance of 0.3 μm. Images were taken at Nyquist to allow sampling at maximum

resolution. Low magnification images to reveal tissue architecture were taken at 1024 × 1024 pixels with Nyquist recommended step size. Line averaging (2X) was used for most images. Images were acquired sequentially in order to avoid bleed-through between channels. Imaging parameters (i.e., laser power, photomultiplier gain, offset, pinhole size, scan speed, etc.) were optimized to prevent pixel saturation and kept constant for all conditions within the same experiment. Images were analyzed using NIS-Elements Advanced Research software (Nikon).

For analysis of synapophysin-2 and gephyrin puncta in tissue, local background subtraction was performed using the rolling ball method. Moreover, to limit variation due to uneven antibody penetration, images were collected consistently at the same edge of the section (e.g. side mounted to the slide). For all other image analysis, background was empirically determined and applied equally to all images from a given imaging session / independent experimental replicate. For quantitative analysis, imaging of brain tissue involved imaging at least 2 regions of interest from 4–5 brain sections. For cultured neurons, two 15–20 μm dendritic segments were analyzed from 8–10 neurons per culture batch, per condition. All ICC/IHC data were collected and analyzed blindly.

For quantifying synaptic puncta, the general analysis module was used in the NIS-Elements Advanced Research software (Nikon). Binary masks were applied to each channel (following background subtraction) for a given image and binary mask settings were optimized and maintained across the images being compared. To look at co-localized puncta, a binary operation was used that conditioned a given mask (e.g. for homer1) as having ("HAVE", the operator) at least one pixel overlap with another mask (e.g. for gephyrin or MAP2). Combinations of binary operations were used to evaluate excitatory and inhibitory synapse density, surface $GABA_AR$ levels, reciprocal and non-reciprocal synapses (gephyrin/vGAT +/- homer1), etc. For synaptic puncta, objects smaller than $0.2\,\mu m^2$ were filtered out. Puncta density was calculated based by dividing the object number by the area of the field of view or length of dendritic segment. Average puncta density was calculated by dividing the binary area by the average number of objects. For most measurements, intensity is included and is the average background-subtracted fluorescence intensity within the area of the assigned binary.

### STED microscopy

STED super-resolution images and confocal comparison images were acquired using a Nikon Ti2-E microscope stand equipped with a STEDYCON confocal and STED module from Abberior Instruments, Inc. Excitation lasers included 488 nm (pulsed), 595 nm (pulsed), and 640 nm (pulsed). A 775 nm STED depletion laser (pulsed) was used. Detection was performed using time-gated APDs. Images were acquired using a CFI PLAN APO LAMBDA 100X OIL objective (NA 1.45) with a piezoelectric focusing system. Immersion oil F was used for all imaging. For quantitative analysis, images were acquired using identical settings. Image settings were optimized to ensure that signal was acquired below saturation, with saturation levels indicated using the look-up table. Acquisition laser intensity was scaled ~1.5X higher for STED imaging and the depletion laser was set to enable 60 nm resolution in all channels. Pixel size was automatically determined based on the resolution of the acquired image. For STED acquisition, 15-line accumulations were used. For analysis, 2 fields of view were collected per section and 4-6 sections per animal. For all images STEDYFOCUS was used to allow sequential confocal and STED imaging. A single optical plane was imaged, as there was no depletion laser in the z plane.

For analysis, OME files were exported from the STEDYCON acquisition software and opened in Hugyen's Essentials from SVI. Express deconvolution, which affords minimal improvement in resolution due to single-plane acquisition, was used to perform background subtraction and images were exported as ASCII files. They were

then opened in Nikon Elements and puncta were analyzed using the general analysis software module. Regardless of fixation duration, the look-up table was scaled to reveal the entire range of signal detected for fitting of binary masks – thus, even under strong fixation with lower signal, we sought to quantify number of puncta even if they were dimmer. Binary mask operations were used to detect dystroglycan (mask 1) that had at least one pixel overlap with gephyrin (mask 2). With STED, although substructure is detectable, masks were dilated to cover the entirety of gephyrin discs, even for large inhibitory synapses with many nanoclusters, which we considered inhibitory post-synapses. Density was determined by dividing the number of identified objects by the area of the field of view. Puncta size was determined by dividing binary area by the total number of objects identified with a given mask.

### Statistics and reproducibility

Quantifications have been described in the respective materials and methods sections, and statistical details are provided in the figure legends and specific p values are described in the Source Data and Statistics table. Statistical significance between various conditions was assessed by determining $p$-values (95% confidence interval). Statistical analyzes were performed using GraphPad Prism 6 or 9 software and Microsoft Excel.

For most staining experiments, the "n" represents the average per animal or average per culture. For qualitative imaging results, experiments were performed at least 3 times. In contrast, for electrophysiology measurements, the "n" represents the total number of cells patched from 3 or more batches of cultures, the number of which is defined in the figures. For biochemical, the "n" generally represents number of animals, independent cultures, or pooled samples. Example images of neurobiotin-filled neurons in Figs. 5b, 8e, 9e, and 6b are included to highlight the infection efficiency and type of neuron being patched, but neurobiotin-filling was not used as a standard for recording experiments. Nevertheless, the infection efficiency and type of neuron being patch clamped were performed on 3+ independent experiments. Most intergroup comparisons were done by two-tailed unpaired $t$ tests with or without Welch's correction. For multiple comparisons, data were analyzed with one- or two-way ANOVA followed by a post-hoc test (e.g. Dunnett's multiple comparison test). Levels of significance were set as $*p < 0.05$; $**p < 0.01$; $***p < 0.001$; $****p < 0.0001$. All graphs depict means ± SEM.

### Reporting summary

Further information on research design is available in the Nature Portfolio Reporting Summary linked to this article.

## Data availability

Source data are provided within this paper. Raw data that support the findings of this study are available from the corresponding author upon request. Source data are provided with this paper.

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

## Acknowledgements

We thank I. Huryeva (Stanford U.) for excellent technical assistance. We would also like to thank Kevin Wright (OHSU) and Jennifer Jahncke (OHSU) for sharing tissue for the initial testing of dystroglycan antibodies for specificity. This study was supported by grants from the NIMH (MH052804 to T.C.S.; KO1-MH105040-01 to J.H.T), a BBRF Young Investigator Grant (to J.H.T.), and a Stanford Interdisciplinary Graduate Fellowship (SIGF) to (C.Y.W.)

## Author contributions

J.H.T., C.Y.W., and T.C.S. designed, and J.H.T., C.Y.W., and P.Z., conducted all experiments. J.H.T. performed all molecular cloning, RNA analysis, and synapse morphology analysis. C.Y.Z. performed all in vivo recordings and P.Z. performed all in vitro recordings. G.N. performed animal perfusions and microscopy. J.H.T., C.Y.W., and T.C.S. analyzed the data and wrote the manuscript; all authors reviewed and approved the final manuscript.

## Competing interests

The authors declare no competing interests.
