## [Peer Review File · Nature Communications]

REVIEWER COMMENTS

Reviewer #1 (Remarks to the Author):

In this study by Trotter et al., the authors assess the role of Neurexin 3 isoform function in vitro and in vivo. Notably, a critical role of Nrnx3-LNS2SS2- in inhibitory synaptic transmission was observed in granule cell to mitral cells synapses in the olfactory bulb and on Layer 5/6 pyramidal neurons in the prefrontal cortex. Using a whole-cell electrophysiology readout, the authors propose that Nrnx3-LNS2SS2- expressed in granule cells interacts transsynaptically with postsynaptic ligand dystroglycan in postsynaptic mitral cells. To test this hypothesis, the authors combine conditional genetic lesions of Nrnx3 and CRISPRi or Cas9-mediated reductions in dystroglycan. Reducing dystroglycan levels in mitral cells produced a phenotype similar to granule cells lacking Nrnx3-LNS2SS2-, supporting a transsynaptic signaling pathway involving dystroglycan and neurexin.

Trotter et al. advance the synaptic organization field by using molecular genetics, viral-mediated lesioning, and whole-cell electrophysiology to assess the role of a functionally uncharacterized Neurexin isoform Nrnx3-LNS2SS2-. Overall this study has the potential to advance the field with some revisions discussed below.

Major comments:

- In the final sentence of section "A combinatorial splice code of SS2 and SS4 controls Nrnx3 function at GCMC synapses in the OB as an 'AND/OR' logic gate", the authors conclude that the combination Nrnx3-SS2- and Nrnx3-SS4- is required to "for Nrnx3 to sustain synaptic function," yet the authors demonstrate that the dominant Nrnx3 isoform in the OB is SS4+ (Figure S3g) which seems to contradict this conclusion. The subsequent use of the and/or boolean logic gate as it applies to this statement is confusing. A diagram of the cascade 'and/or' logic gate as it applies to Nrnx3alpha would be helpful to understand the author's intended application of boolean logic to Neurexin isoform signaling.

- The authors demonstrate a 60% reduction in dystroglycan via CRISPR-based deletion and conclude no reduction in inhibitory synaptic numbers as measured by postsynaptic gephyrin staining. They comment that their results contrast a study by Früh et al., which genetically deleted dystroglycan using NEX-cre and observed a reduction in CCK neuron-pyramidal neuron inhibitory synapses in the hippocampus. It is possible that dystroglycan signaling operates in a cell-dependent manner (OB>MF vs. CCK>pyr), but given the different gene targeting strategies, the authors show a 60% reduction, not a knock-out. Thus, the authors' conclusion that their results contrast the Früh et al. study is too strong of a statement as it is unclear if the authors' inhibitory synapse phenotype would be further aggravated by genetic cKO. Further, the Früh et al. study shows a reduction in inhibitory synapse

numbers via dystroglycan cKO but shows normal numbers of gephyrin puncta. Thus, the authors' use of gephyrin to quantify synapse numbers may be problematic to quantify synapse numbers.

Minor comments:

- In their Introduction, the authors state, "However, only one site of alternative splicing has been shown to be functionally relevant, splice site #4 (SS4), which regulates binding of key ligands to neurexins, and controls the postsynaptic levels of AMPA- (AMPA-Rs) and NMDA-receptors (NMDARs). In a recent publication, Hauser et al. *Neuron* 2022 demonstrate the functional relevance of Neurexin splice site #5 (SS5). Thus, the authors should revise this statement and include Hauser et al. (DOI: 10.1016/j.neuron.2022.04.017)

- In their Summary the authors conclude that Nrnx3alpha-dystroglycan signaling "renders these synapses competent for neurotransmitter release." The authors demonstrate that inhibitory synaptic transmission is reduced/alterd yet still intact and thus still competent for synaptic transmission despite Nrnx3 or dystroglycan deficiency from OB>MF synapses. Thus, the authors should revise this statement.

- In their Summary, the authors conclude that "the Nrnx3 deletion phenotype was again completely rescued by the LNS2-only Nrnx3 construct lacking an insert in SS2." However, in Figure 5e, the authors demonstrate that Nrnx3-LNS2SS2- does not entirely rescue mIPSC amplitude in the mPFC. Thus, the authors should revise this statement.

Reviewer #2 (Remarks to the Author):

The manuscript Trotter, Wang, Zhou et al., is a very accomplished study describing the role of Neurexin-3 in normal release probability at inhibitory synapses via a trans-synaptic feedback loop requiring binding of presynaptic Neurexin 3 to the postsynaptic dystroglycan. The experiments are very well conducted and the figures represent findings in a clear manner. To the best of my knowledge, it is also perhaps the first study showing a direct involvement of dystroglycan in neural transmission. I have the following minor comments:

1. In Fig. 1 SS1 isoform is a bit abruptly introduced, although a reason is described for including it, there is no mention of its expression levels in the OB to show its physiological relevance.

2. The dominant negative effect of Nrnx3b SS4+ is not clearly stated.

3. In Fig. 2 authors have systematically removed different domains to pinpoint which domain and splice variant might be most important for function. Does removing the O-glycosylation stalk of Nrnx3a-LNS2 SS2- have any effect on the synaptic transmission. Since O-glycosylation is important for neural transmission it might be interesting to see if LNS2 SS2- can function properly without the stalk.

4. In Fig 4b-c, the rescue by Nrnx3a-LNS2 SS2- has not been plotted on the cumulative probability plots.

5. In Fig 5. the small decrease in amplitude of miniature IPSCs in mPFC is not rescued by Nrnx3a-LNS2 SS2- and has been attributed to "a different modest role for nrnx3 in postsynaptic GABA(a)R function in the mPFC". GABA(a)R is also present in the GC-MC synapse of the OB and yet there is no decrease in amplitude of the mIPSCs (Fig 4). So, it is not clear what modest role authors are speaking of and is it specific to synapses of mPFC only. Please clarify for a broader audience.

6. In Fig 6 since dystroglycan is only partially inhibited and about 35% of Dag1 is still predicted to be present, does overexpressing Nrnx3a-LNS2 SS2- in these cells rescue the synaptic transmission defect.

Reviewer #3 (Remarks to the Author):

In this paper Trotter et al., study multiples Neurexin-3 alternative splicing, including SS1, SS2 and SS4 using both in vitro and in vivo modelling. In particular, the authors show that selective conformation of Neurexin-3-SS2 and -SS4 can modulate the synaptic release of inhibitory synapses but do not control the number of synapses. The authors, then, show that SS2 splicing is responsible of the binding of Neurexin-3 with dystroglycan. Finally, when they knockdown dystroglycan in OB, the authors show that the lack of dystroglycan disrupts inhibitory synapse function resembling the Neurexin-3 mutants.

Overall, the paper is interesting and well written however multiple points were not properly described and sometimes even omitted throughout the paper. For example, multiple supplementary figures are barely introduced, described and discussed throughout the entire result section. I do understand the difficulty of presenting the massive amount of produced data, but this way the overall impression is that the authors somehow picked and described only a subset of the data. I would also point out that because of all the splicing nomenclature, it is quite complicated to follow the paper, it is extremely difficult to make sense to most of the figures and most of the time even the figure legends are not very useful. Another critical point is that despite the extensive electrophysiological analysis, the authors failed to demonstrate that dystroglycan is located at the synapse of interest. Multiple papers have already mapped out the localization of this singular postsynaptic component and the olfactory bulb has never been associated with this protein before.

Major concerns are here summarized:

Primary mixed olfactory neurons are quite unusual and difficult to obtain. In fact, only limited publication and/or protocols are available. In addition, GC-MC synapses is a very specific and unique type of synapse. Throughout the paper, the authors hint that the primary OB neurons may contain the so called dendrodendritic synapses, however they never even attempt to demonstrate such claim. This type of synapses is unique and, the authors doesn't have any data demonstrating/supporting that a mixed olfactory bulb primary culture can develop such selective synapses. A sentence like "Since granule cells are by far the most abundant inhibitory neurons in the OB, these evoked IPSCs largely reflect GC-MC synaptic transmission in the cultured OB neurons, although they likely also contain a minor component derived from other inhibitory neurons in the cultures" (page 6, lines 122-125) needs to be proved by data. The used OB in vitro model has never been introduced or described before, with the only exception to a 10 year old paper published by the same group. I would highly recommend to better describe, present and characterized the used model or to downplay the GC-MC synapse reference.

On the same line, throughout the entire paper, the authors keep hinting that because GC represents the highest number of cells in the OB (up to 95%) both RT-PCR and ribo-Tag experiments can be transferable to the GC-MC synapses (Figures S3, S4 and more). In addition, in Fig 4e, the authors labelled the graph GC and MC Tag, but it is unclear how they extracted this type of data. Neither the result nor the method section contains an explanation on how the analysis has been executed. There are multiple specific markers for both cell types that could be used for discerning between them. If this is not possible, I would recommend to apply techniques that would allow a better spatial resolution of the cellular type (RNA-scope or similar). I do believe that it is quite important to use one of these two approaches to demonstrate the selective localization of the different SS2 splicing.

Another major limitation of this work is that most of the in vivo manipulations are not cell specific, therefore despite the authors claiming that because HA-Nrx3 is somehow distributed in the same layer (lines 232-234), this approach is specific and suited for further analysis, missing control

experiments are missing. It would also be nice to better described all the used plasmids, and how do they control for their specificity. In addition, it is important to demonstrate that Nr3 is really located at presynaptic compartment and therefore it would be important to demonstrated that the re-expression doesn't affect the Nr3 localization within these synaptic compartments.

Despite the claim of the authors that it is normal to use ROI's rather than number of animals as degree of freedom (lines 240-245) to quantify synapse number, they never described the quantification protocol in the method section. I also would like to point out that multiple valid protocols have already been published and most of them share precise and specific protocols on how to study synapse composition. Because of the immunofluorescence fixation protocol indicated in this paper (most synaptic proteins do not like perfusion), the fact that the authors only use low magnification to quantify synaptic puncta (including homer and gephyrin) and the total lack of a quantification method, I find it quite difficult to really understand and evaluate the synaptic analysis performed in Figure 3.

The result that Neurexin-3 is quite specific for inhibitory synapses in multiple brain regions (OB and mPFC) and potentially other cortical areas, it is quite interesting and important. However, because of the high variety of inhibitory synapses throughout the different brain regions and the unspecific targeting of Neurexin-3 in the mPFC, it would be great if the authors could dissect which cell type most likely contain Neurexin-3 and/or whether this presynaptic component is rather ubiquitously expressed in main inhibitory synapses.

Regarding the dystroglycan, as I already stated, its distribution within the brain has already been described. None of the previous publication reports the expression of this component in the olfactory bulb. Is this therefore mandatory to include a better description of the localization of this complex selectively in the OB and to also characterize with inhibitory synapse most likely contains such complex. Even within the cortical area, only a few cell types have been reported to express this postsynaptic molecule and as mentioned by the authors some of the previous results are still controversial. I also would like to point out that the examples used for showing KD of dystroglycan in (S7a), quite clearly shows an effect on the endothelial dystroglycan rather than an effect on neuronal dystroglycan. It is therefore essential to better control the localization of dystroglycan in the OB.

Finally, I would like to point out that a recent paper published by Briatore et al., 2020 already demonstrated that long-term lack of dystroglycan is crucial for the synaptic organization of cerebellar inhibitory synapses, raising the question whether the authors may have to wait longer to detect the detachment of synapses. If this is the case the electrophysiological phenotype described by the authors could be part of a more complex process, that most likely, starts with impairment of the physiological data but could end up with a macroscopical/anatomical reorganization of the circuits. If possible, I would suggest to study whether the long-term lack of DAG can induce the detachment of inhibitory synapses in the OB (as already described in the cerebellum).

Finally, one of the conclusions of the authors is that Neurexin3 and dystroglycan may bind to each other. In addition of demonstrating that these two components are located in the same synapse, it would also be important to apply biochemistry to conclusively prove their direct binding. Both these proteins are known to bind to extracellular components and possibly to other transmembrane partners. It is therefore possible that the similarity of the phenotype is driven by the disruption of a complex containing both partners rather than a direct binding between these two components

Minor comments

Minor: Figure 1a-b, number of cells recorded is quite different between the two genotypes: 30 vs 16, respectively? Because of the spread distribution of the control cell values, it would be important to understand why the authors doubled the number of analyzed cells in this specific experiment.

Figure 1i: magenta (vGAT) and red (Homer) are almost impossible to distinguish and it is therefore very difficult to differentiate between possible excitatory and inhibitory synaptic puncta. The quantification protocol for this specific experiment is also missing in the method part, so it is unclear what the authors counted for the total inhibitory graph. Please specify it.

Please specify with cortical area has been used for the primary cortical neurons.

Response to the Reviewers' Comments for Trotter et al.

We thank the reviewers for their constructive comments and fair consideration of our paper. To address the reviewers' concerns and suggestions, we have performed extensive additional experiments and made significant textual changes. Specifically, we have performed the following revision experiments that are also described below in our detailed response:

- A. We have performed extensive immunocytochemistry experiments demonstrating that dystroglycan is indeed localized to multiple types of inhibitory synapses in the olfactory bulb and other brain regions (**new Fig. 7 and S8**).
- B. We demonstrate using a specific genetic manipulation of mitral cells that *Nrxn3* acts exclusively presynaptically in granule cell→mitral cell synapses in the olfactory bulb (**new Fig. S6**)
- C. We have now also analyzed synapses using double-labeling for a specific presynaptic marker (synaptoporin) in addition to a postsynaptic marker (gephyrin) to ensure that the results obtained by labeling with only postsynaptic markers are not misleading (**new Fig. S9**)
- D. We have included in situ hybridization data from the Allen Institute to demonstrate that dystroglycan is broadly expressed in neurons throughout the olfactory bulb (OB) and is particularly abundantly present in mitral cells, which are the postsynaptic recipients of the synapses we analyze in the OB (**new Fig. S8**)
- E. Moreover, using single-cell publicly available RNAseq results we show that granule cells account for >80% of the inhibitory neurons in the OB (**new Fig. S3**)

We hope that with the additions and changes made, our paper can be deemed acceptable for publication in *Nature Communications*. In the following, we will cite the reviewers' comments in full in italic typeface and provide our responses in light blue regular typeface.

Reviewer #1

Major comments:

*- In the final sentence of section "A combinatorial splice code of SS2 and SS4 controls *Nrxn3* function at GCMC synapses in the OB as an 'AND/OR' logic gate", the authors conclude that the combination *Nrxn3*-SS2- and *Nrxn3*-SS4- is required to "for *Nrxn3* to sustain synaptic function," yet the authors demonstrate that the dominant *Nrxn3* isoform in the OB is SS4+ (Figure S3g) which seems to contradict this conclusion. The subsequent use of the and/or boolean logic gate as it applies to this statement is confusing. A diagram of the cascade 'and/or' logic gate as it applies to *Nrxn3*alpha would be helpful to understand the author's intended application of boolean logic to Neurexin isoform signaling.*

Agreed. We apologize for the lack of clarity. Our enthusiasm for the observation on how the combinatorial use of alternatively spliced sequences can gate the function of a cell-adhesion molecule ran away with us a bit. We agree with the reviewer's suggestion, and now include a diagram (see Fig. 10) depicting the splice combinations that sustain normal

inhibitory synaptic transmission. In addition, we have improved the text to better reflect this point without evoking Boolean logic.

- The authors demonstrate a 60% reduction in dystroglycan via CRISPR-based deletion and conclude no reduction in inhibitory synaptic numbers as measured by postsynaptic gephyrin staining. They comment that their results contrast a study by Früh et al., which genetically deleted dystroglycan using NEX-cre and observed a reduction in CCK neuron-pyramidal neuron inhibitory synapses in the hippocampus. It is possible that dystroglycan signaling operates in a cell-dependent manner (OB>MF vs. CCK>pyr), but given the different gene targeting strategies, the authors show a 60% reduction, not a knock-out. Thus, the authors' conclusion that their results contrast the Früh et al. study is too strong of a statement as it is unclear if the authors' inhibitory synapse phenotype would be further aggravated by genetic cKO. Further, the Früh et al. study shows a reduction in inhibitory synapse numbers via dystroglycan cKO but shows normal numbers of gephyrin puncta. Thus, the authors' use of gephyrin to quantify synapse numbers may be problematic to quantify synapse numbers.

Agreed. Prompted by the reviewer's correct concerns that gephyrin labeling could bias what we see, we have added additional analyses of synapse numbers using staining for synaptopodin, a specific presynaptic marker of dendro-dendritic synapses (Fig. S9). Similar to our manipulations in Nrnx3 cKO conditions (Fig. 3), we see no changes in synapse numbers as measured by synaptopodin staining following global dystroglycan deletion.

Moreover, we agree that it is possible that the remaining mRNA levels of dystroglycan after the CRISPR-mediated deletion could correspond to functional mRNAs, and that these mRNAs might compensate for a possible dystroglycan function in synapse formation. At the same time, these mRNAs would have to be unable to sustain normal synaptic transmission since the CRISPR-mediated deletion of dystroglycan severely impairs synaptic transmission with a phenotype that is identical with that of the Nrnx3 deletion. According to this scenario, dystroglycan would have to have two synaptic ligands, Nrnx3 for enabling normal synapse function and an unknown ligand for mediating synapse formation. We cannot rule out this hypothesis, but favor the alternative hypothesis, namely that not all cells are infected by the CRISPR viruses, but that in the infected cells dystroglycan is deleted ~100%. This alternative hypothesis is supported by the fact that for the CRISPR manipulations we performed recordings only on mitral / tufted cells that strongly expressed fluorescent reporters, indicating that these cells also have strong expression of the guide RNAs and that the true efficiency of dystroglycan reduction in recorded cells is much higher than 60%.

Please note that several explanations could account for the observation that the deletion of dystroglycan in our experiments do not change synapse numbers, including

- 1: The synapses we are studying in the OB and PFC differ from those studied by others (e.g. CA1 and cerebellum) and don't require dystroglycan to form or be maintained.
2. It is possible that our genetic manipulations occur after a sensitive window in which dystroglycan regulates synapse numbers and/or synapse maintenance.

3. Another possibility is that synapse loss is only evident after prolonged deletion, and thus our relatively narrow postnatal window does not capture this.

Finally and most importantly, the *Nrxn3* deletion is ~100%, and does not cause a loss of synapses, suggesting that for the specific synapses studied, the *Nrxn3*-dystroglycan complex regulates presynaptic release and not synapse numbers.

Minor comments:

- *In their Introduction, the authors state, "However, only one site of alternative splicing has been shown to be functionally relevant, splice site #4 (SS4), which regulates binding of key ligands to neurexins, and controls the postsynaptic levels of AMPA- (AMPA-Rs) and NMDA-receptors (NMDARs). In a recent publication, Hauser et al. Neuron 2022 demonstrate the functional relevance of Neurexin splice site #5 (SS5). Thus, the authors should revise this statement and include Hauser et al. (DOI: 10.1016/j.neuron.2022.04.017)*

Terrific point – we actually also showed previously that SS5 is functionally relevant (Aoto et al., 2015) and apologize for overlooking the Hauser et al. paper. We have revised the wording accordingly, and cited Hauser et al. (2022).

- *In their Summary the authors conclude that *Nrxn3*alpha-dystroglycan signaling "renders these synapses competent for neurotransmitter release." The authors demonstrate that inhibitory synaptic transmission is reduced/altered yet still intact and thus still competent for synaptic transmission despite *Nrxn3* or dystroglycan deficiency from OB>MF synapses. Thus, the authors should revise this statement.*

Agreed – we have re-worded this statement to provide clarity.

- *In their Summary, the authors conclude that "the *Nrxn3* deletion phenotype was again completely rescued by the LNS2-only *Nrxn3* construct lacking an insert in SS2." However, in Figure 5e, the authors demonstrate that *Nrxn3*-LNS2SS2- does not entirely rescue mIPSC amplitude in the mPFC. Thus, the authors should revise this statement.*

Agreed – we have revised the wording accordingly.

Reviewer #2

1. *In Fig. 1 SS1 isoform is a bit abruptly introduced, although a reason is described for including it, there is no mention of its expression levels in the OB to show its physiological relevance.*

We have improved the writing to make the introduction of SS1 less abrupt. Please note that we included SS1 in our analyses because in previous studies on neurexophilins (Wilson et al., 2020), we observed that SS1 modulates the binding of neurexophilins to LNS2 domain. Based on multiple RNAseq studies, SS1 is known to be frequently alternative spliced in brain (e.g., see data from our or from the Scheiffele lab). However, since SS1 had no effect in the synapses we were studying, we did not determine its abundance in the OB.

2. *The dominant negative effect of Nrnx3b SS4+ is not clearly stated.*

Given the low abundance of Nrnx3beta in the olfactory bulb, we do not believe that a dominant negative function is physiologically relevant. We have simplified this statement given space constraints.

3. *In Fig. 2 authors have systematically removed different domains to pinpoint which domain and splice variant might be most important for function. Does removing the O-glycosylation stalk of Nrnx3a-LNS2 SS2- have any effect on the synaptic transmission. Since O-glycosylation is important for neural transmission it might be interesting to see if LNS2 SS2- can function properly without the stalk.*

The current paper focuses on the interaction between dystroglycan and Neurexin-3 that is mediated by the LNS2 domain, which is positioned far away from the stalk domain where extensive O-linked glycosylation and heparan sulfate modifications are present. We agree that O-glycosylation of neurexins is potentially important and plan to address it in follow-up efforts, but this is question is unrelated to the goal of the current study.

4. *In Fig 4b-c, the rescue by Nrnx3a-LNS2 SS2- has not been plotted on the cumulative probability plots.*

The data were actually plotted in the figure, but were difficult to see because the line for Nrnx3a-LNS2 SS2- is almost identical to the line for delta-Cre. We have changed the thickness of Nrnx3a-LNS2 SS2- line so that both lines are clearer. Zooming in is still required, as the lines closely resemble one another.

5. *In Fig 5. the small decrease in amplitude of miniature IPSCs in mPFC is not rescued by Nrnx3a-LNS2 SS2- and has been attributed to "a different modest role for nrnx3 in postsynaptic GABA(a)R function in the mPFC". GABA(a)R is also present in the GC-MC synapse of the OB and yet there is no decrease in amplitude of the mIPSCs (Fig 4). So, it is not clear what modest role authors are speaking of and is it specific to synapses of mPFC only. Please clarify for a broader audience.*

Agreed - we have included a better description of this observation and its interpretation in the results section. In brief, other neurexins in the olfactory bulb are likely sufficient to compensate for the absence of Nrnx3 as it relates to GABAAR function. As is often the case in synapses throughout the brain, individual Nrnx's have diverse functions that are likely underwritten by the individual expression levels, pattern of alternative splicing, and ligand availability. The attraction of the OB synapses that form the main focus of our paper is that they are relatively homogeneous whereas the inhibitory synapses in the mPFC are comprised of a population of synapse types with different properties and distinct neurexins and neurexin ligands.

6. *In Fig 6 since dystroglycan is only partially inhibited and about 35% of Dag1 is still predicted to be present, does overexpressing Nrnx3a-LNS2 SS2- in these cells rescue the synaptic transmission defect.*

We have focused our rescue efforts on *in vivo* experiments because they are physiologically relevant and have not performed additional experiments in culture. Moreover, given that we only patched bright report-positive cells infected to express CRISPRi KRAB-dCas9 / gRNA and that not all cells were GFP-positive, including glia and some neurons, we predict the actual level of reduced expression in patched neurons is higher. The phenotypic similarity of *in vitro* and *vivo* manipulations of Nr3n3 (cKO and rescues) and DAG1, argue in favor of this notion. Nevertheless, we would be happy to repeat the *in vivo* rescue experiments in cultured neurons if the reviewers and editors felt this would enhance our conclusions, but such experiments are labor intensive and time consuming, and would take a minimum of 6 months.

Reviewer #3 (Remarks to the Author)

We numbered this reviewer's comments below to enable us to cross-reference them.

1. Overall, the paper is interesting and well written however multiple points were not properly described and sometimes even omitted throughout the paper. For example, multiple supplementary figures are barely introduced, described and discussed throughout the entire result section. I do understand the difficulty of presenting the massive amount of produced data, but this way the overall impression is that the authors somehow picked and described only a subset of the data.

We thank the reviewer for acknowledging the interest of our paper and apologize that we did not go into sufficient depth in describing our extensive data, which are as the reviewer stated quite substantial. We have tried to improve this problem in the revised paper, but as is often the case, we must also adhere to size limits. Inclusion of voluminous data, much of it from control experiments, reflects our commitment to not cherry-picking data that should be shown, as would be the case by leaving it out. The figures do not exclude any data, nor do we intentionally describe only a subset of the data.

In writing this paper, we were caught between two opposite goals: On the one hand, we would prefer to publish a single substantial paper that is complete, instead of multiple less voluminous papers that each focuses on a different single message and showcases only a subset of findings. On the other hand, we would like to be able to communicate the main conclusion of our study as clearly and convincingly as we can without distracting readers by additional, less important findings. To improve achieving both goals, we have now in the revised manuscript described more data in depth and have tried to be more clear about the additional conclusions that our results enable without distracting from the main message.

2. I would also point out that because of all the splicing nomenclature, it is quite complicated to follow the paper, it is extremely difficult to make sense to most of the figures and most of the time even the figure legends are not very useful.

Again, we apologize for a lack of clarity. The reviewer is correct that alternative splicing adds an additional layer of complexity to our understanding of a gene's function, and that we and others who routinely analyze the effects of alternative splicing tend to use an often arcane terminology of alternative splicing. We have now added schematics explaining the

alternative splicing of *Nrxn3* to Fig. 1, and have included a summary diagram in Fig. 10 that explains the splice code of *Nrxn3* as it relates to dystroglycan.

3. Another critical point is that despite the extensive electrophysiological analysis, the authors failed to demonstrate that dystroglycan is located at the synapse of interest. Multiple papers have already mapped out the localization of this singular postsynaptic component and the olfactory bulb has never been associated with this protein before.

We agree that localizing dystroglycan to OB synapses is an important additional line of evidence in support of our findings, although several elegant previous studies have already done this. In particular, prior work by Zaccaria et al. (2001) reported prominent expression of dystroglycan by mitral cells, and to a lesser extent, tufted cells. Using immuno-EM, they found a striking postsynaptic expression of dystroglycan within reciprocal synapses in the external plexiform layer of the OB, data that we now confirm.

To further address this issue, we have performed additional immunostaining experiments of dystroglycan in the olfactory bulb using the extensively validated anti-alpha-dystroglycan IIH6C4 monoclonal antibody (Früh et al., 2016; Miller et al., 2021; Uezu et al., 2019). As frequently required for synaptic proteins (Gasser et al., 2006), we found that dystroglycan labeling could only be observed after light fixation (new Fig. 7, S8). The improved detection of dystroglycan and gephyrin allowed us to confirm with super-resolution microscopy (two dimensional STED) that inhibitory synapses in the EPL contain dystroglycan, and that a subset of large synapses in the glomeruli also have dystroglycan (new Fig. 7). Moreover, we confirmed with the same antibody the expected expression profiles of dystroglycan in the hippocampus and cerebellum. These data clearly document that in addition to the prominent presence of dystroglycan in the wall of blood vessels, dystroglycan is abundantly present in a subset of inhibitory synapses, including the reciprocal synapses in the OB that we analyzed.

Major concerns are here summarized:

4. Primary mixed olfactory neurons are quite unusual and difficult to obtain. In fact, only limited publication and/or protocols are available. In addition, GC-MC synapses is a very specific and unique type of synapse. Throughout the paper, the authors hint that the primary OB neurons may contain the so called dendrodendritic synapses, however they never even attempt to demonstrate such claim. This type of synapses is unique and, the authors doesn't have any data demonstrating/supporting that a mixed olfactory bulb primary culture can develop such selective synapses.

We agree that dissociated mixed neuron-glia cultures from OB are not commonly studied, but we have published such cultures in multiple prior studies (Cao et al., Cell 2011 and J. Neurosci. 2013; Aoto et al., 2015). We are aware of dendrodendritic synapses in the OB since we studied them extensively (Liu et al., Neuron 2017; Wang et al., Neuron 2020; Wang et al., Sci. Advances 2021 in addition to the papers cited above). The data shown in our various papers and the present submission shows that the OB culture system presents several advantages over the widely used hippocampal and cortical cultures. In particular, the high ratio of smaller granule cells to larger mitral/tufted cells that can be

easily morphologically identified allows us to reliably study granule cell→mitral cell synapses. By comparison, hippocampal and cortical cultures are rather heterogeneous. Our original papers (cited above) contain extensive protocols on OB cultures, but to meet the reviewer's concerns we have now in the current paper included additional detail to our culture protocol. In the materials and methods, we layout the exact steps required to culture dissociated olfactory bulb neurons. We hope that this will make this culture method more accessible to general readers.

However, we do not concur with the reviewer's statement that we '*never even attempt to demonstrate*' that these cultures form dendrodendritic synapses. In our paper, we quantified putative reciprocal synapses in Figure S2 based on the adjacency of vGAT- and gephyrin-positive inhibitory puncta to Homer1-positive excitatory puncta that are co-localized on MAP2-positive dendrites. To provide further clarity, we have now relocated some of these quantifications to Figure 1 and highlight putative reciprocal synapses in representative images in Figure S2. By this measure, the majority of inhibitory synapses on mitral/tufted cells in OB cultures were reciprocal synapses. Super-resolution microscopy clearly demonstrated the adjacent localization of excitatory and inhibitory specializations in putative reciprocal dendrodendritic synapses (Fig. 7). We did not see a change in either Homer1-positive or Homer1-negative inhibitory synapse numbers after either the *Nrxn3* or the dystroglycan deletions (Fig. 1, S2).

5. A sentence like "Since granule cells are by far the most abundant inhibitory neurons in the OB, these evoked IPSCs largely reflect GC-MC synaptic transmission in the cultured OB neurons, although they likely also contain a minor component derived from other inhibitory neurons in the cultures" (page 6, lines 122-125) needs to be proved by data.

We agree, and have now confirmed the commonly held view that the majority of inhibitory neurons in the OB are granule cells by analyzing single-cell RNAseq data from the OB that allow quantification of the abundance of different types of neurons (Tempe et al., 2017). Our new data, included in the revised Figure S3, demonstrate that ~83% of inhibitory neurons in the OB are granule cells, supporting the statement. Given the uncertainty about the exact proportion of GC→MC synapses and their relative contribution to synaptic currents in culture, we have extensively re-worded interpretation of culture data to reflect inhibitory neuron → MC synapses. Indeed, the fact that our *in vivo* findings resemble the culture data argues that this synapse is likely also severely impaired in culture. Nevertheless, we can't rule out impairment in other inhibitory synapses in culture and did not specifically study other inhibitory synapses onto MCs in slice.

6. The used OB *in vitro* model has never been introduced or described before, with the only exception to a 10 year old paper published by the same group. I would highly recommend to better describe, present and characterized the used model or to downplay the GC-MC synapse reference.

Please see our response to comment 4, explaining that this culture system has already been used in multiple other papers but that we have now expanded the description in the current paper.

7. On the same line, throughout the entire paper, the authors keep hinting that because GC represents the highest number of cells in the OB (up to 95%) both RT-PCR and ribo-Tag experiments can be transferable to the GC-MC synapses (Figures S3, S4 and more). In addition, in Fig 4e, the authors labelled the graph GC and MC Tag, but it is unclear how they extracted this type of data. Neither the result nor the method section contains an explanation on how the analysis has been executed. There are multiple specific markers for both cell types that could be used for discerning between them. If this is not possible, I would recommend to apply techniques that would allow a better spatial resolution of the cellular type (RNA-scope or similar). I do believe that it is quite important to use one of these two approaches to demonstrate the selective localization of the different SS2 splicing.

We agree that the mRNAs isolated via the vGAT(+)-Cre/Ribotag method contains a significant contribution from inhibitory neurons other than granule cells, since ~17% of inhibitory neurons in the OB are not granule cells (see comment 5 above). We have now amended the text to reflect this finding. This, however, does not change the overall interpretation of our data because it suggests that not only in granule cells, but also in other inhibitory neurons of the OB, SS2 in *Nrxn3* is nearly completely spliced out. In other words, all GABAergic neurons of the OB lack an insert in SS2 for *Nrxn3*. As an aside, it would not be possible to examine SS2 by RNA-scope because its sequence is too short, and current single-cell RNAseq techniques would generally find it difficult to analyze neurexin alternative splicing at SS2 because the sequencing depth is too shallow.

8. Another major limitation of this work is that most of the in vivo manipulations are not cell specific, therefore despite the authors claiming that because HA-Nrx3 is somehow distributed in the same layer (lines 232-234), this approach is specific and suited for further analysis, missing control experiments are missing.

The reviewer is correct that the majority of the *Nrxn3* manipulations are not cell specific, whereas the *in vivo* deletions of dystroglycan are. For the dystroglycan deletions, we used both mitral/tufted cell Cre mice (i.e. Tbet-Cre) and retro-AAV-Cre, to allow targeted postsynaptic deletion of DAG1 in the presence of conditional cas9 alleles. The cell-type specificity for all experiments derives from the electrophysiological recording and imaging strategy that enables exclusive analysis of granule cell→mitral cell synapses.

To the best of our knowledge, no tools are available for specific targeting of granule cells in the OB. However, to assuage the reviewer's concerns that we have not confirmed that *Nrxn3* acts presynaptically since the genetic deletions are not cell-type specific, we performed additional recordings on mitral cells from *Nrxn3* cKO mice in which *Nrxn3* is only deleted postsynaptically in mitral cells using retro-AAV's that were injected into the piriform cortex. Recordings were only performed on infected mitral cells. Unlike the global deletion of *Nrxn3*, the selective postsynaptic deletion of *Nrxn3* in mitral cells did not impair granule cell→mitral cell synapses (Figure S6).

9. It would also be nice to better described all the used plasmids, and how do they control for their specificity.

Agreed - in a supplemental file, we now include unpublished sequences of the plasmids used and/or overviews of their design. All viral rescue constructs were driven by human synapsin-1 promoter for efficient expression in neurons, which have been used by other labs in addition to our own in hundreds of paper. Moreover, as is the case for all papers from our lab, all published plasmids are shared freely upon request.

10. In addition, it is important to demonstrate that Nr3 is really located at presynaptic compartment and therefore it would be important to demonstrated that the re-expression doesn't affect the Nr3 localization within these synaptic compartments.

We don't quite understand the rationale behind this comment. Neurexins have been shown to act exclusively presynaptically in many papers, including papers from our own lab (e.g., see Aoto et al., 2013 and 2015; Dai et al., 2021 and 2022; Luo et al., 2019). Moreover, Wang et al. (Sci. Advances, 2021) recently documented that neurexins are also presynaptic in dendrodendritic synapses in the OB. Our revision experiments that show postsynaptic deletion of Nr3 in mitral cells does not change GC→MC synaptic transmission is consistent with this. We would like to ask the reviewer to explain why it would be necessary to document this again as this has been published.

11. Despite the claim of the authors that it is normal to use ROI's rather than number of animals as degree of freedom (lines 240-245) to quantify synapse number, they never described the quantification protocol in the method section. I also would like to point out that multiple valid protocols have already been published and most of them share precise and specific protocols on how to study synapse composition. Because of the immunofluorescence fixation protocol indicated in this paper (most synaptic proteins do not like perfusion), the fact that the authors only use low magnification to quantify synaptic puncta (including homer and gephyrin) and the total lack of a quantification method, I find it quite difficult to really understand and evaluate the synaptic analysis performed in Figure 3.

We believe that the reviewer misunderstood us here and have revised the manuscript to be clearer. Our main point in showing averages per animal is to highlight a more rigorous form of image analysis that contrasts an approach of analyses which treats individual images (neurons, sections, ROIs, etc.) as independent experimental observations and statistical replicates. We refer to these as pseudoreplicates. Our concern is that pseudoreplicates can lead to spurious conclusions. Indeed, in our hands pseudo-replicates show statistically significant effects that are driven by variation between individual animals (Fig. 3). To actually claim there is an effect, which other cross-validating measures don't support, many more animals would have to be studied.

Regarding methods for quantifying synapses, we respectfully disagree with the reviewer. There are actually few, if any, widely accepted methods and practices for quantifying synapse numbers in tissue. It is generally accepted that double-labeling for pre- and

postsynaptic markers is superior to single labeling, but how to assess synapse numbers is not generally agreed. Although we have taken a stance against pseudoreplication for image analysis in this paper, that is obviously not a major goal of this submission.

The reviewer also appears to misunderstand our protocol for staining, imaging, and analysis. We have added additional detail to our materials and methods for clarity. It is important to note that we did use light fixation (i.e. 7 minutes perfusion with 4% PFA followed by < 20 minute post-fixation) for our synaptic labeling as described in the Methods.

Finally, we believe that the reviewer additionally misunderstood how we analyze synapse numbers here (and in many other published studies) because he/she/they states “*the authors only use low magnification to quantify synaptic puncta*”. Synapse analyses were only performed at high magnification, as described in the paper. The low magnification images are shown to provide a field view of a section or culture, but the actual quantifications were obviously performed with the high magnification images that are also shown. The high magnification images were acquired at Nyquist and have the highest possible resolution for confocal microscopy. The confusion here could result from the large fields that we still include at high magnification – we did this so as to not show “cherry-picked” puncta, but if the reviewer desires, we can include even more zoomed in images. We believe that the current images – upon zooming in – are sufficient to reveal synaptic puncta that were quantified.

12. The result that Neurexin-3 is quite specific for inhibitory synapses in multiple brain regions (OB and mPFC) and potentially other cortical areas, it is quite interesting and important. However, because of the high variety of inhibitory synapses throughout the different brain regions and the unspecific targeting of Neurexin-3 in the mPFC, it would be great if the authors could dissect which cell type most likely contain Neurexin-3 and/or whether this presynaptic component is rather ubiquitously expressed in main inhibitory synapses.

Again, this is a complete misunderstanding of the current paper and earlier papers from our and other groups. Nowhere do we suggest that *Nrxn3* is specific for inhibitory synapses. Quite the opposite. For example, we previously showed that *Nrxn3* in hippocampal and cerebellar excitatory synapses controls AMPARs in a manner regulated by alternative splicing at SS4 (Aoto et al., Cell 2013; Dai et al., Nature 2020 and E-Life, 2021). If this confusion arises from the recent paper by Hauser et al. (2022), it is important to note that their tag knockin localizes only *Nrxn3* isoforms that contain a specific SS5 variant. Across the brain, this splice site is absent in ~80-90% of transcripts, and thus the distribution pattern that they observed doesn't remotely encompass overall *Nrxn3* distribution (Treutlein and Gokce et al., 2014; Schreiner et al., 2014). In actuality, all neurons likely express all neurexins at different ratios (Ullrich et al., 1995). Specificity arises not from the expression of different neurexins, but from their alternative splicing and from the complement of ligands that are expressed in the postsynaptic targets of a neuron. As it related to excitatory synaptic function, in the current manuscript, this was only studied in olfactory cultures, and we found no change in excitatory synapse number or function (recording from MCs) in *Nrxn3* cKO and following CRISPRi of DAG1. For this

reason, our *in vivo* recordings exclusively focused on inhibitory synaptic transmission for structure-function experiments in culture and for all *in vivo* manipulations. Given the major contribution of Nrnx3 to SS2(-) transcripts to in OB and elsewhere (given its strong preference for exclusion), we predict that its absence might be less easily compensated for in the context of synapses in which Nrnxns and dystroglycan interact.

13. Regarding the dystroglycan, as I already stated, its distribution within the brain has already been described. None of the previous publication reports the expression of this component in the olfactory bulb. Is this therefore mandatory to include a better description of the localization of this complex selectively in the OB and to also characterize with inhibitory synapse most likely contains such complex. Even within the cortical area, only a few cell types have been reported to express this postsynaptic molecule and as mentioned by the authors some of the previous results are still controversial. I also would like to point out that the examples used for showing KD of dystroglycan in (S7a), quite clearly shows an effect on the endothelial dystroglycan rather than an effect on neuronal dystroglycan. It is therefore essential to better control the localization of dystroglycan in the OB.

Again we would like to respectfully disagree with the reviewer's assessment here. Expression of dystroglycan is widespread in brain and has been previously described in the OB. In addition to the work by Zaccaria et al. (see our response to comment 3), the Allen Brain Atlas as well as innumerable RNAseq studies document abundant expression of dystroglycan in neurons throughout the brain, including the OB. To make this point clearer, we have now included *in situ* hybridization data from the Allen Institute in Figure S8 that document the abundance of dystroglycan mRNA in the OB, particularly in the mitral cell layer. These findings are consistent with our functional recordings that support a postsynaptic role of dystroglycan at inhibitory synapses formed upon mitral cells. Moreover, we have previously described the mRNA distribution of all 6 principal neurexins (Nrnx1-3 α and Nrnx1-3 β) in the mouse olfactory bulb using *in situ* hybridization (Ullrich et al., 1995). We have also included in the current manuscript *in situ* hybridization results from the Allen Institute for both Nrnx3 and DAG1 (Fig. S8). Immunostaining of neurexins, especially individual neurexins, has been hampered by a lack of sufficiently specific antibodies. Several years ago, we reported on a HA-Nrnx1 cKI mouse (Trotter et al., 2019), but we have not generated and validated a Nrnx3 tag knockin mouse yet.

14. Finally, I would like to point out that a recent paper published by Briatore et al., 2020 already demonstrated that long-term lack of dystroglycan is crucial for the synaptic organization of cerebellar inhibitory synapses, raising the question whether the authors may have to wait longer to detect the detachment of synapses. If this is the case the electrophysiological phenotype described by the authors could be part of a more complex process, that most likely, starts with impairment of the physiological data but could end up with a macroscopical/anatomical reorganization of the circuits. If possible, I would suggest to study whether the long-term lack of DAG can induce the detachment of inhibitory synapses in the OB (as already described in the cerebellum).

We agree that a synaptic dysfunction as we observe in our experiments could lead to a secondary loss of synapses. Indeed, it would be interesting to investigate if synaptic dysfunction necessarily leads to synapse elimination, and to explore the molecular mechanisms involved. We also agree that one would expect an impairment in a subset of synapses to lead to a reorganization of the affected neural circuits, certainly by homeostatic plasticity and possibly by other mechanisms as well. However, investigating these questions is beyond the scope of the current study, which is focused on the control of inhibitory synapses by trans-synaptic Nrnx3-dystroglycan signaling regulated by alternative splicing. Although outside the scope of the current study, we have included additional discussion related to this point in the manuscript.

15. Finally, one of the conclusions of the authors is that Neurexin3 and dystroglycan may bind to each other. In addition of demonstrating that these two components are located in the same synapse, it would also be important to apply biochemistry to conclusively prove their direct binding. Both these proteins are known to bind to extracellular components and possibly to other transmembrane partners. It is therefore possible that the similarity of the phenotype is driven by the disruption of a complex containing both partners rather than a direct binding between these two components

This experiment has previously been published (Sugita et al., 2001; Reissner et al., 2014). The biochemical definition of the direct neurexin-dystroglycan complex and its regulation by alternative splicing at SS2 motivated the current project, as described throughout the present manuscript. Importantly, our functional findings are 100% consistent with the binding data from these earlier studies, both of which lacked functional analysis.

Minor comments

13. Minor: Figure 1a-b, number of cells recorded is quite different between the two genotypes: 30 vs 16, respectively? Because of the spread distribution of the control cell values, it would be important to understand why the authors doubled the number of analyzed cells in this specific experiment.

This experiment has 8 groups. mIPSC recordings are more time consuming than evoked responses. For this reason, 3 separate batches of cultures were recorded from for dCre and dCre + rescue constructs. Thus, dCre had a total of 6 groups. This separation was necessary to ensure that enough cells could be analyzed during the DIV14-DIV17 time window.

14. Figure 1i: magenta (vGAT) and red (Homer) are almost impossible to distinguish and it is therefore very difficult to differentiate between possible excitatory and inhibitory synaptic puncta. The quantification protocol for this specific experiment is also missing in the method part, so it is unclear what the authors counted for the total inhibitory graph. Please specify it.

We have updated the Figure 1i and S2 to include new colors and have included a better description in the methods for how synapses were quantified.

15. Please specific with cortical area has been used for the primary cortical neurons.

The entire cortical lobes were used following their dissection from midbrain, hindbrain, and the hippocampus. The materials and methods have been updated to reflect this.

We hope that with the additional experiments and the corrections in the manuscript, the paper can now be deemed acceptable for publication, and thank the reviewers for their careful assessment of our study.

REVIEWERS' COMMENTS

Reviewer #1 (Remarks to the Author):

In the revised study by Trotter et al., the authors improve several aspects of their investigation into the roles of Neurexin 3 synaptic function in the olfactory bulb and mPFC. The combinatorial logic of neurexin splicing is inherently complicated and the revised manuscript is easier to read. The addition of super-resolution imaging of DAG enhances the paper and is impressive. Overall, I am satisfied with the authors' responses to my comments, and this second read of the manuscript has significantly improved the paper. I recommend for publication.

Reviewer #2 (Remarks to the Author):

The authors have addressed all my concerns and they do not need to repeat the *in vivo* rescue experiments in cultured neurons. The revised manuscript is now acceptable for publication.

Reviewer #3 (Remarks to the Author):

The authors' rebuttal exhaustively responded to all my concerns and accordingly revised the paper. I do not have additional comments.

Response to the Reviewers' Comments for Trotter et al.

We thank the reviewers for their comments and fair consideration during the initial review of our paper. Moreover, we thank the reviewers for their overwhelmingly positive reviews of the revised manuscript, which do not require a detailed response. We look forward to having this important body of work shared with the broader community.